# Exploration in the Face of Strategic Responses: Provable Learning of Online Stackelberg Games

## Abstract

We study online leader-follower games where the leader interacts with a myopic follower using a quantal response policy. The leader's objective is to design an algorithm without prior knowledge of her reward function or the state transition dynamics. Crucially, the leader also lacks insight into the follower's reward function and realized rewards, posing a significant challenge. To address this, the leader must learn the follower's quantal response mapping solely through strategic interactions — announcing policies and observing responses. We introduce a unified algorithm, *Planning after Estimation*, which updates the leader's policies in a two-step approach. In particular, we first jointly estimate the leader's value function and the follower's response mapping by maximizing the sum of the Bellman error of the value function, the likelihood of the quantal response model, and a regularization term that encourages exploration. The leader's policy is then updated through a greedy planning step based on these estimates. Our algorithm achieves a sublinear regret in the context of general function approximation. Moreover, this algorithm avoids the intractable optimistic planning and thus enhances implementation simplicity.

## 1 Introduction

Stackelberg games are a class of games that feature strategic decision-making under a leader-follower structure. These games find broad applications in various domains, such as economics, finance, societal systems, and so on (He et al., 2007; Von Stackelberg, 2010; Keyhani, 2003; Sinha et al., 2013; Ghosh & De, 2021; Koh et al., 2020; Qiu et al., 2021). In the simplest two-player case, the two players are referred to as the *leader* and *follower*, respectively. These two players have misaligned objectives and different information structures, and their interactions can be sequential and dynamic.

In this game, the leader has more advantages in the sense that she can regularize the follower's behavior by announcing her policy before the two players take actions and promising to commit to it. In that case, the leader's policy becomes common knowledge. The follower, knowing the leader's policy, determines his policy by solving his decision-making problem determined by both the leader's policy and the follower's reward function. As a result, the follower's policy is a strategic response to the leader's policy, and such a mapping (the response model) depends on the follower's reward function. From the leader's point of view, the response model specifies how the followers strategically interact with the leader, and the leader aims to maximize her cumulative rewards in expectation. The leader's policy that maximizes her cumulative rewards in the presence of the strategic follower, together with the follower's response policy, constitutes a Stackelberg equilibrium of the game. This notion characterizes the optimal behavior of such a leader-follower game.

While there have been many existing works proposing sample-efficient multi-agent reinforcement learning (MARL) algorithms for solving dynamic games, the study of solving Stackelberg equilibria from data via MARL is relatively scarce. Most of these works focus on Nash-(Perolat et al., 2017), Correlated-(Cigler & Faltings, 2011), or coarse correlated equilibria(Sessa et al., 2022) of Markov games. When it comes to Stackelberg equilibria, the hierarchical and strategic nature make it hard to learn from data. The main challenge lies in the estimation of the response model of the follower. When the response model is unknown to the leader, she needs to infer the response model, or equivalently, estimate the follower's reward function from data. This entails a challenging exploration problem – the leader has to find a sequence of policies such that the follower's responses to them are sufficiently

informative. Moreover, as shown in Bai et al. (2021), such a problem is ill-posed when the follower is fully rational, i.e., returning a deterministic reward-maximizing action. In this case, even if the follower's reward function is accurately estimated, the resulting estimated response model still has a considerable error.

To address this challenge, Chen et al. (2023) propose to study Markov Stackelberg games (MSG) with the follower adopting a quantal response model. That is, the follower solves an entropy-regularized reward maximization problem and the response policy is stochastic. In this case, after announcing a policy and observing the action taken by the follower, the leader can estimate the follower's reward function via maximum likelihood estimation (MLE). Based on this observation, in the online setting, Chen et al. (2023) proposes a sample-efficient algorithm based on optimistic planning, in the context of general function approximation. In particular, their algorithm constructs a confidence for the follower's reward function via MLE, and a confidence set for the leader's value function using the Bellman error. However, due to the hierarchical structure, this Bellman error takes the follower's reward function as a parameter. As a result, optimistic planning over these coupled confidence sets is highly intractable. Therefore, the following question remains elusive:

> *Can we design a sample-efficient and easy-to-implement MARL framework*
> *for Markov Stackelberg games with general function approximation?*

In this paper, we provide an affirmative answer to this question. Focusing on the online setting of Markov Stackelberg games where the follower is myopic and boundedly rational, we propose an easy-to-implement algorithm, dubbed Planning after Estimation (PES). In particular, in each episode, the algorithm updates the leader's policy in two steps. First, in the estimation step, we estimate the leader's value function and the follower's quantal response model together using a combined loss function. This loss function combines (i) the likelihood loss for estimating the follower's reward function, (ii) the Bellman loss for estimating the leader's value function, and (iii) an additional term that promotes exploration. Such an exploration-promoting term is defined as the expected rewards of the leader based on the given value function and response model. In the second step, based on the estimated value function and response model, we update the leader's policy by solving the greedy policy. Compared to the optimistic planning algorithm proposed in Chen et al. (2023), our algorithm circumvents intractable optimistic planning, which involves joint planning of the leader's policy, value function, and the follower's quantal response model. Furthermore, we prove that PES achieves a sublinear $\tilde{O}(d_c \sqrt{T})$-regret with general function approximation, where $d_c$ is the decoupling coefficient (Xiong et al., 2022) that captures the complexity of the employed function classes and $T$ is the number of episodes. As a result, our PES is provably sample efficient and amenable to implementation at the same time. Furthermore, as a concrete example, we instantiate the leader-follower game to the problem of reinforcement learning with human feedback (RLHF), demonstrating the efficacy of our algorithm.

## 2 RELATED WORK

**Online Stackelberg Games.** Most existing works on learning Stackelberg equilibria in (Markov) games via online RL assume the follower is myopic and perfectly rational (Bai et al., 2021; Zhong et al., 2023; Kao et al., 2022; Zhao et al., 2023). In specific, Bai et al. (2021); Zhao et al. (2023) focus on the static setting. Bai et al. (2021) consider a centralized setting where the central controller can determine the actions taken by both the leader and the follower, and Zhao et al. (2023) assume the follower is omniscient in the sense that the follower always plays the best response policy, which is similar to our setting. They show that when the follower is perfectly rational, the regret of the leader exhibits different scenarios depending on the relationship between the leader's and the follower's rewards. Besides, Kao et al. (2022) assume that the leader and follower are cooperative and design a decentralized algorithm for both the leader and follower, under the tabular setting. Zhong et al. (2023) study online and offline RL for the leader, assuming the follower's reward function is known, and thus the best response of the follower is known to the leader. Our work is more related and comparable to Chen et al. (2023). In particular, Chen et al. (2023) extensively studied Markov Stackelberg games in the context of general function approximation. They proposed an algorithm framework, which is provably sample efficient under assumptions that the follower is bounded rational and either myopic or farsighted. However, they constructed confidence sets for the response model and leader's value function and introduced optimistic planning (Auer et al., 2008) to update the leader's policies.

Such a method involves joint planning of the leader's policy and value function, and the follower's quantal response model, so the planning steps become computationally intractable, which means the algorithm is very hard to be implemented in practice. In this paper, we propose our PES algorithm to overcome this drawback. Instead of using tedious optimistic planning, we exploit the benign property of the Shannon entropy function to recover the follower's reward function via his policy. After estimating the reward function, we execute the planning step by solving the "greedy" policy. Compared with Chen et al. (2023), our algorithm is not only easy-to-implement but also easier to show theoretical guarantee.

**Online RL with General Function Approximation.** Recently, various works propose RL algorithms in the context of general function approximation (Jiang et al., 2017; Sun et al., 2019; Jin et al., 2021; Xiong et al., 2022; Liu et al., 2024a). Among these works, Our work is most relevant to Jin et al. (2021); Xiong et al. (2022); Liu et al. (2024a). Specifically, Jin et al. (2021); Xiong et al. (2022) introduce the Multi-agent decoupling coefficient that characterizes the exploration difficulty of the Markov Decision Process (MDP) problems. In Section 5, we introduce similar notions of decoupling coefficient for learning the leader's optimal policy. In particular, we introduce two versions of the decoupling coefficient that capture the complexity of the leader's Bellman error and the follower's quantal response error. Besides, Liu et al. (2024a) proposed an easy-to-implement RL algorithm framework named Maximize to Explore (MEX) and instantiating MEX on the 2-player zero-sum game setting. However, their algorithm framework can not be easily instantiated in MSG, because the follower's Bellman error is not accessible in our setting, since either the follower's reward function or his realized rewards remains unknown.

# 3 PRELIMINARIES

**Notation** For a measurable space $\mathcal{X}$, we use $\Delta(\mathcal{X})$ to denote the set of probability measure on $\mathcal{X}$. For an integer $n \in \mathbb{N}$, we use $[n]$ to denote the set $\{1, ..., n\}$. For a random variable $X$, we use $\mathbb{E}[X]$ and $\text{Var}[X]$ to denote its expectation and variance respectively. For two functions $f(x)$ and $g(x)$, we denote $f(x) = O(g(x))$ if there is a constant $C$ s.t. $f(x) \leq C \cdot g(x), \forall x \in \text{Dom}(f) \cap \text{Dom}(g)$ and we use $\tilde{O}(\cdot)$ to omit all the logarithmic terms. For two functions $f, g : \mathcal{A} \to \mathbb{R}$, we denote $\langle f, g \rangle_{\mathcal{A}} = \sum_{a \in \mathcal{A}} f(a) \cdot g(a)$.

## 3.1 LEADER-FOLLOWER MARKOV GAMES

**Problem Settings.** A leader-follower Markov Game is between two players, referred to as the leader and the follower, respectively (also called principal and agent in other literature). These two leaders will interact within an episode of $H$ steps and the states of the game evolve according to a Markov transition kernel. Let $\mathcal{S}$ be the state space, and let $\mathcal{A}$ and $\mathcal{B}$ be the action sets of the leader and follower, respectively. Let $P = \{P_h : \mathcal{S} \times \mathcal{A} \times \mathcal{B} \to \Delta(\mathcal{S})\}_{h \in [H]}$ denotes the transition kernels of the $H$ steps, and let $u = \{u_h : \mathcal{S} \times \mathcal{A} \times \mathcal{B} \to [0,1]\}_{h \in [H]}$ and $r = \{r_h : \mathcal{S} \times \mathcal{A} \times \mathcal{B} \to [0,1]\}_{h \in [H]}$ be the leader and follower's reward functions of $H$ steps, respectively.

In contrast to a classic Markov game, the leader-follower game features an additional "communication stage": before the beginning of the game, where the leader announces a policy $\pi = \{\pi_h : \mathcal{S} \to \Delta(\mathcal{A})\}_{h \in [H]}$ and the follower adopts a response policy $\upsilon^{\pi} = \{\upsilon_h^{\pi} : \mathcal{S} \to \Delta(B)\}_{h \in [H]}$ according to a response model: $\pi \to \upsilon^{\pi}$. Then the two players play the joint policy $(\pi, \upsilon^{\pi})$ and generate a trajectory $\{s_h, a_h, b_h\}_{h \in [H]}$. In particular, at step any step $h \in [H]$, the leader and follower observe the current state $s_h \in \mathcal{S}$, take actions $a_h \sim \pi_h(\cdot|s_h)$ and $b_h \sim \upsilon_h^{\pi}(\cdot|s_h)$, receive rewards $u_h(s_h, a_h, b_h)$ and $r_h(s_h, a_h, b_h)$ respectively, and the environment moves to a new state $s_{h+1} \sim P_h(\cdot|s_h, a_h, b_h)$. Here, we could assume the initial state $s_1$ is sampled from a fixed distribution $\rho_0 \in \Delta(\mathcal{S})$ and the game terminates after $s_{H+1}$ is generated. At last, we define $\Pi = \Pi_1 \times \Pi_2 \times \cdots \Pi_H$, where $\Pi_h = \Delta(\mathcal{A})$, as the domain of the leader's policy $\pi$.

**Quantal Response Model.** In the above discussion, we mentioned that after the leader announces its policy $\pi$, the follower will choose its policy $\upsilon^{\pi}$ according to this context. We feature this process as quantal response models: $\pi \to \upsilon^{\pi}$. In this paper, we mainly discuss the boundedly rational and myopic follower, where the "myopic" means the follower only tries to maximize his expected immediate reward and the "boundedly rational" means the follower considers other factors (represented as a regularization term) when maximizing his rewards. We define the quantal response

policy of the follower with respect to $\pi$, denoted by $\upsilon^\pi$ as the solution to an entropy regularized policy optimization problem:

$$\upsilon_h^\pi(\cdot|s) = \arg\max_{\nu_h} \left\{ \mathbb{E}^{\pi_h,\nu_h}\left[r_h(s_h, a_h, b_h)|s_h = s\right] + \frac{1}{\eta}\mathcal{H}(\nu_h(\cdot|s))\right\}, \forall s \in \mathcal{S}, \qquad (3.1)$$

where $\mathbb{E}^{\pi_h,\nu_h}[\cdot|s_h = s]$ means we take expectation with respect to $(\pi_h, \nu_h)$. Here $\mathcal{H}$ is a strongly convex regularization function and $\eta > 0$ is a parameter. In order to solve the corresponding problem directly and give a closed-form solution of the follower's policy in equation (3.1), we further assume the regularization function $\mathcal{H}$ is the Shannon entropy of the follower's policy. Here we don't rule out the possibility of using other regularization functions, and consider such extension as our future work.

**Stackelberg Equilibrium.** When the follower adopts the response model $\upsilon$, the goal of the leader is to find the optimal $\pi^\star$ that maximizes its expected total rewards when the follower's response model is $\upsilon^\pi$, i.e.,

$$\pi^\star \in \arg\max_\pi J(\pi), \qquad J(\pi) = \mathbb{E}^\pi\left[\sum_{h\in[H]} u_h(s_h, a_h, b_h)\right]. \qquad (3.2)$$

Here $\mathbb{E}^\pi$ denotes the expectation over the trajectory $\{s_h, a_h, b_h\}_{[h\in H]}$ generated by the joint policy $(\pi, \upsilon^\pi, P)$ and the maximization in (3.2) is over all policies of the leader. The optimal leader's policy $\pi^\star$ and its response $\upsilon^{\pi^\star}$ constitutes a Stackelberg (Markov perfect) equilibrium. That is, Stackelberg equilibrium characterizes the leader's optimal policy, when the follower adopts a particular response model that maps each a leader's policy $\pi$ to a follower's policy $\upsilon^\pi$.

## 3.2 Online Stackelberg Game

In this paper, we consider the learning problem of the leader in the online setting. That is, without any prior knowledge about the reward functions $u$ and $r$ and transition model $P$, the leader aims to learn $\pi^\star$ by repeatedly playing the same game with a follower and adaptively gathering data, where the follower adopts the response model $\upsilon^\pi$. Specifically, the leader adaptively constructs a sequence of policies $\{\pi^t\}_{t\geq 1}$ where $\pi^t$ is the policy in the $t$-th episode. The leader's data consists of the trajectories and bandit feedback of the follower's reward generated by playing the game. In particular, when leader adopts $\pi^t$, the follower adopts $\upsilon^{\pi^t}$ and they generate a trajectory $\{s_h^t, a_h^t, b_h^t\}_{h\in[H]}$. The leader observes this trajectory as well the bandit feedback of her reward, i.e., $\{u_h(s_h^t, a_h^t, b_h^t)\}_{h\in[H]}$. Based on the data generated before the $t$-th episode, the leader constructs $\pi^t$ by a learning algorithm and uses it to generate new data. Here a key assumption of our setting is that the leader does not know the follower's realized rewards or the reward function, which is realistic but also the source of the major technical challenge.

To evaluate the performance of the learning algorithm, we use the notion of sample complexity. Let $\epsilon \in (0, 1)$ be the desired error level, the sample complexity is defined as the smallest integer $T_\epsilon$ such that the algorithm constructs an $\epsilon$-optimal policy $\hat{\pi}$ after $T_\epsilon$ episodes, where $\hat{\pi}$ satisfies $J(\pi^\star) - J(\hat{\pi}) \leq \epsilon$. Specifically, the performance is measured by the regret, which is as

$$\text{Reg}(T) = \sum_{t=1}^T \left(J(\pi^*) - J(\pi^t)\right). \qquad (3.3)$$

# 4 Algorithm Framework: Planning after Estimation

## 4.1 Formulate Stackelberg Games via Reinforcement Learning

**Formulate Stackelberg games into Bilevel Optimization.** First, we recall the quantal response policy of the follower $\upsilon^\pi$ could be viewed as the solution to an entropy regularized policy optimization problem, which is given by equation (3.1). Thus, we could write the maximization of the leader's reward as a bilevel optimization problem:

$$\pi = \arg\max_{\pi\in\Pi^H} J(\pi), \qquad J(\pi) = \mathbb{E}^\pi\left[\sum_{h\in[H]} u_h(s_h, a_h, b_h)\right], \qquad (4.1)$$

$$\upsilon_h^\pi(\cdot|s) = \arg\max_{\nu_h} G_h(\pi, \nu), \qquad G_h(\pi, \nu) = \left\{\mathbb{E}^{\pi_h,\nu_h}\left[r_h(s, a_h, b_h)|s\right] + 1/\eta \cdot \mathcal{H}(\nu_h(\cdot|s))\right\}, \forall s \in \mathcal{S},$$

where the leader's problem is in the upper level: for each leader's policy $\pi$, we find the follower's optimal policy $\upsilon^\pi$ induced by $\pi$, and then find the optimal $\pi^\star$ that maximizes the leader's reward.

**Leader's Value Functions.** Let $U_h^\pi : \mathcal{S} \times \mathcal{A} \times B \to \mathbb{R}$ and $W_h^\pi : \mathcal{S} \to \mathcal{B}$ to be the leader's action-value (U) function and state-value (W) function under policy $\pi$, which are defined as:

$$U_h^\pi(s_h, a_h, b_h) = u_h(s_h, a_h, b_h) + \mathbb{E}^P \left[ W_{h+1}^\pi(s_{h+1}) | s_h, a_h, b_h \right]$$
$$= u_h(s_h, a_h, b_h) + (P_h W_{h+1}^\pi)(s_h, a_h, b_h), \tag{4.2}$$

$$W_h^\pi(s_h) = \mathbb{E}^{\pi, \upsilon^\pi} [U_h^\pi(s_h, a_h, b_h)] = \left\langle U_h^\pi(s_h, \cdot, \cdot), \pi_h \otimes \upsilon_h^\pi(\cdot, \cdot | s_h) \right\rangle_{\mathcal{A} \times \mathcal{B}}, \tag{4.3}$$

where the expectation in equation (4.3) is taken w.r.t. $a_h \sim \pi_h(\cdot | s_h, b_h)$, $b_h \sim \upsilon_h^\pi(\cdot | s_h)$. Here we define $P_h W(s, a, b) = \sum_{s' \in \mathcal{S}} P_h(s' | s, a, b) W(s')$ and define $\pi_h \otimes \upsilon_h^\pi(a, b | s) = \pi_h(a | s, b) \cdot \upsilon_h^\pi(b | s)$ for $\forall h \in [H]$.

Intuitively, $U_h^\pi$ and $W_h^\pi$ are counterparts of the $Q$-function and $V$-function in standard RL, respectively. That is, $W_h^\pi(s)$ is equal to the expected total rewards starting from $s_h = s$ and the two players follow $\pi$ and $\upsilon^\pi$. Thus, the total reward of the leader is given by $J(\pi) = \mathbb{E}_{s_1 \sim \rho_0}[W_1^\pi(s_1)]$, where $\rho_0$ is the initial state distribution. For simplicity, we consider the case when the initial state $s_1$ is fixed.

**Bellman Equation** By (4.1), we notice that, since the follower is myopic, his response policy at each step $h$ could be computed separately, which means $\upsilon_h^\pi$ depends on $\pi$ only through $\pi_h$. As a result, to find the optimal policy $\pi^\star$, it only suffices to optimize $\pi_h$ at each step separately in (4.3), which leads to the following Bellman optimal equation for $\{U_h^\star, W_h^\star, \pi^\star\}_{h \in [H]}$:

$$U_h^\star(s_h, a_h, b_h) = u_h(s_h, a_h, b_h) + P_h W_{h+1}^\star(s_h, a_h, b_h),$$
$$W_h^\star(s_h) = \max_{\pi_h(\cdot | s_h) \in \Delta(\mathcal{A})} \left\{ \left\langle U_h^\star(s_h, \cdot, \cdot), \pi_h \otimes \upsilon_h^\pi(\cdot, \cdot | s_h) \right\rangle \right\}, \tag{4.4}$$

and $\pi_h^\star$ is the optimal policy that achieves the maximum in (4.4). In other words, $\pi^\star$ is the "greedy" policy with respect to $U^\star$ and the quantal response mapping $\pi \to \upsilon^\pi$.

**Recovering Standard MDPs.** A special case of this leader-follower game is when $\mathcal{B}$ is a singleton. In this situation, this game reduces to a standard MDP, because $b_h = b$ is fixed and $\upsilon^\pi$ degenerates into $\delta(b)$ for any $\pi \in \Pi$. Then we can recover the classical Bellman equation in (4.4). Thus, estimating $u$ and $P$ is the same as in standard RL.

## 4.2 Model Estimation and Greedy Planning

According to the discussion in Section 4.1, we know that once the leader knows reward functions $u$ and $r$, and the transition kernel $P$, then they can solve equation (4.4) to find the optimal policy $\pi^\star$. In the online setting, we need to approximately solve equation (4.4) using online data $\{\pi^t, \{s_h^t, a_h^t, b_h^t, u_h^t\}_{h \in [H]}\}_{t \in [T]}$. From the leader's perspective, there are two types of unknown quantities: (i) the follower's response model $\upsilon^\pi$, which depends on the follower's reward function $r = \{r_h\}_{h \in H}$; (ii) the leader's reward function $u = \{u_h\}_{h \in [H]}$ and the transition kernel $P = \{P_h\}_{h \in H}$.

**Estimate the Response Model.** To deal with the unknown quantities relevant to the follower, we need to consider how his policy $\upsilon^\pi$ involves with leader's value function $U, W$. A natural way to tackle this is via estimating the response model $\pi \to \upsilon^\pi$. In general, it is an intractable task since it means we need to estimate a functional mapping from $\Delta(\mathcal{A})$ to $\Delta(\mathcal{B})$. A promising approach to address this challenge is to directly estimate the follower's reward function through his announced policy, instead of estimating the complicated response model. This approach is equivalent to solving an *inverse optimization problem* $\upsilon^\pi$ shown in equation (4.1). That is, given a solution (the follower's policy) of the optimization problem, could we recover the parameter (the follower's reward function) of this problem? This kind of inverse problem is usually ill-posed. However, thanks to the benign property of the Shannon entropy, we could write the follower's policy in closed form:

$$\upsilon_h^\pi(b_h | s_h) = \exp \left( \eta \cdot (A_h^\pi(s_h, b_h)) \right), \qquad A_h^\pi(s_h, b_h) = r_h^\pi(s_h, b_h) - V_h^\pi(s_h), \tag{4.5}$$
$$r_h^\pi(s, b) = \mathbb{E}_{a \sim \pi}[r(s, a, b)] = \langle \pi_h(\cdot | s, b), r_h(s, \cdot, b) \rangle_{\mathcal{A}},$$
$$V_h^\pi(s_h) = 1/\eta \cdot \log \left( \sum_{b \in \mathcal{B}} \exp(\eta \cdot r_h^\pi(s_h, b)) \right).$$

Here $V_h^\pi(s)$ is a normalizing constant ensuring $\upsilon_h^\pi(\cdot | s) \in \Delta(\mathcal{B})$, and $\eta > 0$ is a parameter. By this closed-form solution, we know the inverse optimization problem of estimating $r$ is well-posed and

can be solved simply via maximum likelihood estimation (MLE). In particular, we can view $v^\pi$ as a statistical model with parameter $r$, and equation (4.5) enables us to compute the likelihood function when observing data $\{\pi_h^t, s_h^t, b_h^t\}_{t \in [T]}$.

To this end, we approximate $r$ using a function class $\mathcal{F}_r = \{r^\theta : \mathcal{S} \times \mathcal{A} \times \mathcal{B} \to [0, 1]\}_{\theta \in \Theta}$, where $\theta$ is a parameter with respect to $r$ and $\Theta$ is the the parameter space. We further assume the realizability condition on reward function $r$:

**Assumption 1** ((Realizability of Reward Function)). *There exists $\theta^\star \in \Theta$ such that $r^{\theta^\star} = r$.*

Then, we can estimate $r_h^\star = r_h^{\theta^\star}$ by using the negative log-likelihood loss:

$$L_{h,1}^t(\theta_h) = -\sum_{i=1}^{t-1} \log v_h^{\pi^i, \theta}(b_h^i \mid s_h^i) = -\sum_{i=1}^{t-1} \eta \cdot A_h^{\pi^i, \theta}(s_h^i, b_h^i). \tag{4.6}$$

where $v_h^{\pi, \theta_h}$ and $A_h^{\pi, \theta_h}$ are defined in equation (4.5) with $r_h$ replaced by $r_h^{\theta_h}$. Let $\theta = \{\theta_h\}_{h \in [H]}$, we further define $v^{\pi, \theta} = \{v_h^{\pi_h, \theta_h}\}_{h \in [H]}$.

**Estimate the Value Function.** To deal with the unknown quantities relevant to the leader, we try to estimate her value function. We let $\theta^\star = \{\theta_h^\star\}_{h \in [H]}$ where $\theta_h^\star$ is the parameter of $r_h$, then we have:

$$U_h^\star(s_h, a_h, b_h) = u_h(s_h, a_h, b_h) + P_h W_{h+1}^\star(s_h, a_h, b_h),$$

$$W_h^\star(s_h) = \max_{\pi_h(\cdot|s_h) \in \Delta(\mathcal{A})} \left\{ \left\langle U_h^\star(s_h, \cdot, \cdot), \pi_h \otimes v_h^{\pi, \theta^\star}(\cdot, \cdot|s_h) \right\rangle \right\},$$

and $W_{h+1}^\star$ appears in the above Bellman equation. As a result, instead of estimating $u_h$ alone, we aim to estimate $U_h^\star = u_h + P_h W_{h+1}^\star$, which is known as the Bellman target in online RL. The estimation of this target is well-studied in the literature. We can either use model-based or model-free approaches. In this paper, we exploit the model-free approach to minimize our assumptions on the function class of the leader's reward function and transition kernel.

To this end, we approximate $U^\star$ using a function class $\mathcal{U} = \mathcal{U}_1 \times \mathcal{U}_2 \times \cdots \times \mathcal{U}_H$, where $\mathcal{U}_h \subset (\mathcal{S} \times \mathcal{A} \times \mathcal{B} \to \mathbb{R})$. To simplify our discussion, we introduce two types of Bellman operator $\{\mathbb{T}_h^{\star, \theta}\}_{h \in [H], \theta \in \Theta}$, which is common in the literature (Perolat et al., 2015; Jin et al., 2022):

$$\mathbb{T}_h^{*, \theta} U(s_h, a_h, b_h) = u_h(s_h, a_h, b_h) + \mathbb{E}_{s_{h+1} \sim P_h(\cdot|s_h, a_h, b_h)}[(T_{h+1}^{*, \theta} U_{h+1})(s_{h+1})],$$

$$\text{where } T_h^{*, \theta}(U_h)(s_h) = \max_{\pi_h \in \Delta(\mathcal{A})} \left\langle U_h(s_h, \cdot, \cdot), \pi_h \otimes v_h^{\pi, \theta}(\cdot, \cdot \mid s_h) \right\rangle.$$

The corresponding Bellman error is defined as:

$$l_h^i(U_h', U_{h+1}, \theta)(s_h^i, a_h^i, b_h^i, s_{h+1}^i) = (U_h' - u_h)(s_h^i, a_h^i, b_h^i) - T_{h+1}^{*, \theta} U_{h+1}(s_{h+1}^i). \tag{4.7}$$

Then, we estimate $U_h^\star$ by minimizing the Bellman error, and the loss function is defined as

$$L_{h,2}^t(U, \theta) = \sum_{i=1}^{t-1} l_h^i(U_h, U_{h+1}, \theta_{h+1})^2 - \inf_{U' \in \mathcal{U}_h} \sum_{i=1}^{t-1} l_h^i(U', U_{h+1}, \theta_{h+1})^2. \tag{4.8}$$

**Greedy Planning after Estimation.** After identifying the loss functions we use to bound the two types of unknown quantities that are mentioned in Section 4.2, we propose Planning after Estimation (PES, Algorithm 1) for solving online Stackelberg Games in the context of general function approximations. We first give a generic algorithm framework and then compare our algorithm with other concurrent works.

---

**Algorithm 1** Planning after Estimation (PES)

---

1: **Initial**: $\mathcal{D} = \emptyset$.
2: **for** $t = 1, 2, \cdots, T$ **do**
3:     Calculate $U^t, \theta^t = \arg\max_{U, \theta} \left( W_1^{U, \theta}(s_1) - \eta_1 \sum_{h=1}^H L_{h,1}^t(\theta_h) - \eta_2 \sum_{h=1}^H L_{h,2}^t(U_h, \theta_h) \right)$.
4:     Execute $\pi^t = \arg\max_{\pi \in \Pi} \left\langle U_1^t(s_1, \cdot, \cdot), \pi_1 \otimes v_1^{\pi, \theta^t}(\cdot, \cdot \mid s_1) \right\rangle_{\mathcal{A} \times \mathcal{B}}$.
5:     Collect data $D_t = \{D_h^t\}_{h \in [H]}$ with $D_h^t = (s_h^t, a_h^t, b_h^t, u_h^t, \pi_h^t)$, and update $\mathcal{D} = \mathcal{D} \cup D_t$.
6: **end for**

---

In each episode $t \in [T]$, the agent first estimates the value function $U^t$ and reward function $\theta^t$ using historical data $\{D^s\}_{s \in [t-1]}$ by maximizing a composite objective given in Algorithm 1. Specifically, in order to achieve exploiting history knowledge while encouraging exploration, the agent considers the composite objective that sums: (a) the negative log-likelihood loss $L_{h,1}^t(\theta^t)$, which represents the exploitation of the agent's current knowledge of the follower's policy; (b) the Bellman error $L_{h,2}^t(U^t, \theta^t)$, which represents the exploitation of the agent's current knowledge on the Bellman target; (c) the expected total return of the optimal policy associated with our chosen $(U^t, \theta^t)$, i.e., $W_1^{U^t, \theta^t}$, which represents exploration for a higher return. With tuning parameters $\eta_1, \eta_2$, the agent balances the weight put on the tasks of exploitation and exploration.

Then the agent predicts $\pi^t$ via the optimal policy associated with the solved $(U^t, \theta^t)$, execute $\pi^t$ to collect data $D_t = \{(s_h^t, a_h^t, b_h^t, u_h^t, \pi_h^t)\}_{h \in [H]}$, and update the loss function $L_{h,1}^t, L_{h,2}^t$.

**Comparison with Optimistic Planning (Chen et al., 2023)** The algorithm proposed in Chen et al. (2023) first built a confidence set $C_{\mathcal{U}, \Theta}$ for $(U^\star, \theta^\star)$, and then predicts $\pi$ via optimal policy with solved $(U^t, \theta^t)$. We could formulate their estimation and planning steps as:

$$(U^t, \theta^t) = \underset{(U, \theta) \in C_{\mathcal{U}, \Theta}(\beta)}{\arg \max} \left\langle U_1(s_1, \cdot, \cdot), \pi_1 \otimes \upsilon_1^{\pi, \theta}(\cdot, \cdot \mid s_1)) \right\rangle_{\mathcal{A} \times \mathcal{B}},$$

$$\pi^t(s_h) = \underset{\pi \in [\Delta(\mathcal{A})]^H}{\arg \max} \left\langle U_1^t(s_1, \cdot, \cdot), \pi_1 \otimes \upsilon_1^{\pi, \theta^t}(\cdot, \cdot \mid s_1) \right\rangle_{\mathcal{A} \times \mathcal{B}}.$$

The most important difference between our PES algorithm and their optimistic planning algorithm is that they need to solve a *constrained optimization* problem inside the complicated confidence set, which is often intractable in practice. The reason is that the confidence set $C_{\mathcal{U}, \Theta}$ is coupled. Intuitively, it can be written as

$$C_{\mathcal{U}, \Theta} = \{(U, \theta) \colon \theta \in C_\Theta, U \in C_{\mathcal{U}}(\theta)\}.$$

Here $C_\Theta$ a confidence set for $\theta^\star$, constructed by the MLE loss in (4.6) for estimating $\theta^\star$, and $C_{\mathcal{U}}(\theta)$ is a confidence for $U^\star$ based on the Bellman error in (4.7), which involves a parameter $\theta$. Instead, PES only needs to maximize a composite objective, i.e. solve an *unconstrained optimization* problem, which is not only tractable but also easy to implement in practice.

We need to highlight that PES is not a Lagrangian duality of the constrained optimization objectives within data-dependent level-sets proposed by Chen et al. (2023) or any other optimistic planning algorithm that could potentially solve this task. In fact, PES could fix the parameter choice $\eta_1, \eta_2$ across each episode $t$. Thus $\eta_1, \eta_2$ is independent of data and predetermined, which contrasts Lagrangian methods that involve an inner loop of optimization for the dual variables.

## 5 REGRET ANALYSIS FOR PES ALGORITHM

### 5.1 GENERAL FUNCTION APPROXIMATION

It is well known that RL with function approximation is intractable without any further assumptions (Krishnamurthy et al., 2016; Weisz et al., 2021). Therefore, it is common to make additional assumptions over the function class in the literature on general function approximation in MDPs, especially for the realizability and completeness assumptions (Wang et al., 2020; Jin et al., 2021; Dann et al., 2021).

**Value Function Approximation.** As MSG could be seen as an extension of MDPs, the generalized realizability and completeness assumptions are also adopted in this work.

**Assumption 2** (Realizability of Value Function). *For the Stackelberg equilibrium, it holds that* $U_h^\star \in \mathcal{U}_h$. *Moreover, for any* $\pi \in \Pi$ *and any* $\theta \in \Theta$, *it holds that* $U_h^{\pi, \theta} \in \mathcal{U}_h$, *where we define*

$$U_h^{\pi, \theta}(s, a, b) = u_h(s, a, b) + P_h W_{h+1}^{\pi, \theta}, \qquad W_h^{\pi, \theta}(s) = \left\langle U_h^{\pi, \theta}(s, \cdot, \cdot), \pi_h \otimes \upsilon_h^{\pi, \theta}(\cdot, \cdot \mid s) \right\rangle_{\mathcal{A} \times \mathcal{B}}.$$

**Assumption 3** (Completeness of Value Function). *For any* $U \in \mathcal{U}, \pi \in \Pi$, *and* $\theta \in \Theta$, *we have* $\mathbb{T}_h^{\pi, \theta} U \in \mathcal{U}_h$. *That is, the Bellman operator* $\mathbb{T}_h^{\pi, \theta}$ *is closed with respect to* $\mathcal{U}$.

In the previous subsection, we introduced the Bellman errors in equation (4.7). However, our regret analysis is more related to the *squared* Bellman errors. Such phenomena have been well studied in the literature of single-agent setting (Dann et al., 2021) and multiple-agent setting (Xiong et al., 2022). Following Xiong et al. (2022), here we define the decoupling coefficient to capture the hardness of our learning problem.

**Definition 1** (Multi-agent Decoupling Coefficient). *Given a two-player Stackelberg Game $\mathcal{M}$, a function class $\mathcal{F}$ and a set of probability measure $\varrho$, the decoupling coefficient $d(\mathcal{M}, \mathcal{F}, \varrho)$ is the smallest real number $d$ such that for any $\mu > 0$ and any $\{\rho^t\}_{t \in [T]} \subseteq \varrho$, we have*

$$\sum_{h=1}^{H} \sum_{t=1}^{T} \mathbb{E}^{\rho^t} [f^t(s_h, a_h, b_h)] \leq \mu \cdot \sum_{h=1}^{H} \sum_{t=1}^{T} \sum_{i=1}^{t-1} \mathbb{E}^{\rho^i} \left[ (f^t(s_h, a_h, b_h))^2 \right] + \frac{d}{4\mu}.$$

Then we identify the Bellman residual class $G_L = \{U_h - \mathbb{T}_h^{*,\theta} U_{h+1}, U_h \in \mathcal{U}_h, U_{h+1} \in \mathcal{U}_{h+1}, \theta \in \Theta, h \in [H]\}$ and the decoupling coefficient $d_1 = d(\mathcal{M}, G_L, \varrho_1)$ to capture the complexity of leader's Bellman error, where

$$\varrho_1 = \{\rho \in \Delta(\mathcal{S} \times \mathcal{A} \times \mathcal{B}) : \rho = \mathbb{P}^\pi((s_h, a_h, b_h) = (\cdot, \cdot, \cdot))\}.$$

is the measure set generated by any $(\pi, \upsilon^\pi, P), \forall \pi \in \Pi$.

**Follower's Policy Approximation.** Here we still use the decoupling coefficient to capture the complexity of quantal response error. We identify the reward residual class $G_F = \{r_h^\theta - r_h, \theta \in \Theta, h \in [H]\}$ and the decoupling coefficient $d_2 = d(\mathcal{M}, G_F, \varrho_2)$, where

$$\varrho_2 = \{\mathbb{P}^\pi(a_h = \cdot \mid \cdot, \cdot)\delta_{(s_h, b_h)}(\cdot, \cdot) - \mathbb{P}^\pi((a_h, b_h) = (\cdot, \cdot) \mid \cdot)\delta_{(s_h)}(\cdot), \forall \pi \in \Pi\},$$

and we define $\mathbb{P}^\pi(a_h, b_h \mid s_h) = \mathbb{P}^\pi(a_h \mid b_h, s_h) \cdot \upsilon_h^\pi(b_h \mid s_h)$, and $\delta_{(s_h, b_h)}$ is the probability measure that assigns 1 to the pair $(s_h, b_h)$.

To simplify the notation, we denote the integral operator $\mathcal{T}_h^\pi$ as

$$\mathcal{T}_h^\pi(r)(s_h, b_h) = \langle \pi_h(\cdot \mid s_h, b_h), r(s_h, \cdot, b_h) \rangle - \langle \pi_h \otimes \upsilon_h^\pi(\cdot, \cdot \mid s_h), r(s_h, \cdot, \cdot) \rangle. \tag{5.1}$$

Then by the definition of $\varrho_2$, for any $\pi \in \Pi, h \in [H]$, there exists one probability measure $\rho \in \varrho_2$ such that $\mathcal{T}_h^\pi(r)(s_h, b_h) = \mathbb{E}^\rho[r(s_h, b_h)]$ and vice versa.

## 5.2 BOUNDS FOR THE DECOUPLING COEFFICIENT.

Here we provide several examples whose decoupling coefficient is provably small.

**Linear MSG** The first example is the MSG with linear function approximation, which is generalized from the definition of linear Markov Game in Xie et al. (2020)

**Definition 2** (Linear MSG). *We say a Markov Stackelberg game is linear, if there exists a feature map $\phi(s, a, b) \in \mathbb{R}^d$ such that for any $(s, a, b) \in \mathcal{S} \times \mathcal{A} \times \mathcal{B}$, $s' \in \mathcal{S}$, and $h \in [H]$, it holds that $u_h(s, a, b) = \phi(s, a, b)^\top \varphi_h^\star$, $P_h(s' \mid s, a, b) = \phi(s, a, b)^\top \mu_h(s')$ and $r_h(s, a, b) = \phi(s, a, b)^\top \theta_h^\star$, for some unknown $\varphi_h^\star, \mu_h(\cdot), \theta_h^\star \in \mathbb{R}^d$ satisfying $\max\{\|\theta_h^\star\|, \|\mu_h\|, \|\varphi_h^\star\|\} \leq \sqrt{d}$*

We have the following upper bound for the decoupling coefficient:

**Proposition 1.** *For a $d$-dimensional MSG with the function class $\mathcal{U}_h = \{(\phi_h^\top \varphi_h) : \|\varphi_h\| \leq (H - h + 1)\sqrt{d}\}$ and $\mathcal{F}_h^r = \{(\phi_h^\top \theta_h) : \|\theta_h\| \leq \sqrt{d}\}$ and $\|\phi(s, a, b)\| \leq 1$, $\forall(s, a, b) \in \mathcal{S} \times \mathcal{A} \times \mathcal{B}$, then we have*

$$d_1, d_2 \leq 2dH \cdot (2 + \ln(2HT)).$$

**Generalized Linear MSG** We consider a MSG with generalized linear function approximation.

**Definition 3** (Generalized Linear MSG). *We say an MSG is generalized linear, if there exists a feature map $\phi(s, a, b) \in \mathbb{R}^d$ such that for any $(s, a, b) \in \mathcal{S} \times \mathcal{A} \times \mathcal{B}$, $s' \in \mathcal{S}$, and $h \in [H]$, it holds that $u_h(s, a, b) = \sigma(\phi(s, a, b)^\top \varphi_h^\star)$, $P_h(s' \mid s, a, b) = \sigma(\phi(s, a, b)^\top \mu_h(s'))$ and $r_h(s, a, b) = \sigma(\phi(s, a, b)^\top \theta_h^\star)$, for some unknown $\varphi_h^\star, \mu_h(\cdot), \theta_h^\star \in \mathbb{R}^d$, where $\sigma$ is differentiable and strictly increasing. We further assume that $\sigma' \in (c_1, c_2)$ for some $c_1, c_2 \in \mathbb{R}$.*

**Proposition 2.** *For a $d$-dimensional MSG with the function class $\mathcal{U}_h = \{\sigma(\phi_h^\top \varphi_h) : \|\varphi_h\| \leq (H - h + 1)\sqrt{d}\}$ and $\mathcal{F}_h^r = \{\sigma(\phi_h^\top \theta_h) : \|\theta_h\| \leq \sqrt{d}\}$ and $\|\phi(s, a, b)\| \leq 1$, $\forall(s, a, b) \in \mathcal{S} \times \mathcal{A} \times \mathcal{B}$, then we have*

$$d_1, d_2 \leq 2 \cdot c_2^2/c_1^2 \cdot dH \cdot (2 + \ln(2HT)).$$

## 5.3 Theoretical Guarantee.

Then we can get the following theorem for the PES algorithm.

**Theorem 1.** *If we choose $\eta_1 = \eta_2 = 1/\sqrt{T}$, then for any $\delta \in (0, 1/3)$ with probability at least $1 - 3\delta$, the Algorithm 1 achieves a regret*

$$Reg(T) \leq \left( H(\beta_1 + \beta_2) + 4C_\eta^2 d_1 + 16(C_0 + C_1)^2 d_2 \right) \sqrt{T} + O(H \log(H/\delta)), \tag{5.2}$$

*where $C_\eta = O(\eta^{-1} + B_A)$, $B_A = 2(\eta^{-1} \log(|\mathcal{B}|) + 1)$, $C_1 = \eta^2 \exp(2\eta B_A)(2 + \eta B_A \cdot \exp(2\eta B_A))/2$, and $\beta_1$ and $\beta_2$ are defined in Lemma 7 and 8 respectively.*

Here we provide a proof sketch for Theorem 1 and defer the detailed proof in Appendix B.

**Step 1.** At first, we decompose the regret into two terms: one is from the estimation error between $(U^\star, \theta^\star)$ and $(U^t, \theta^t)$; the other is from the approximation error when we execute the greedy policy to generate $\pi^t$, i.e. the difference between $W^{U^t, \theta^t}$ and $W^{\pi^t}$.

**Step 2.** To bound the estimation error, we first notice that $(U^t, \theta^t)$ maximizes of the loss function defined in Algorithm 1. Thus, we could upper bound this error with the difference between the loss functions. By lemma 7 and 8, we could bound the difference of $L_{h,1}^t$ and $L_{h,2}^t$, respectively.

**Step 3.** To bound the approximation error, we introduce the performance difference lemma proposed by Chen et al. (2023), which decompose the approximation error into the expected Bellman residuals and the expected estimation error of the follower's policy.

**Step 4.** By the decoupling coefficient assumption, we could transfer the errors that we get in Step 3 into terms relevant to Step 2. By choosing the right kind of $\eta_1, \eta_2$, we could get the regret bound.

The main difference between our algorithm and other concurrent works is that our algorithm not only circumvents the intractable optimistic planning, but also achieves $\tilde{O}(\sqrt{T})$-regret guarantee with simplest hyper-parameter choice: $\eta_1 = \eta_2 = 1/\sqrt{T}$, which means our algorithm is easy-to-implement and does not need to tune or search the best hyper-parameter.

# 6 Case Study: Reinforcement Learning with Human Feedback

Our algorithm can also be applied to the Reinforcement Learning with Human Feedback (RLHF) setting by formulating the RLHF as a turn-based Stackelberg game. Specifically, given the initial distribution $\rho$ and the prompt $x \sim \rho$, the Large Language Model (leader) generates two outputs $a = (y_1, y_2)$ as the action, and the human agent's (follower) action is binary, $y_1 \succ y_2$ or $y_1 \prec y_2$, indicating which output the human prefers. We denote $b = 1$ if $y_1 \succ y_2$ and $b = 0$ if $y_1 \prec y_2$. Finally, the leader observes the human's preference and collects the data $(x, a, b)$. Define the reward function $R(x, y) \in [0, 1]$ over the outputs, and the leader's and follower's reward functions are given by

$$u(x, a, b) = R(x, y_1) + R(x, y_2), \forall b \in \{0, 1\}.$$
$$r(x, a, b = 1) = R(x, y_1) - R(x, y_2),$$
$$r(x, a, b = 0) = R(x, y_2) - R(x, y_1).$$

We can simplify the notation $u(x, a, b)$ as $u(x, a)$ since it is not dependent on the preference $b$. Using the reward model above, the quantal response of the follower is given by

$$\mathbb{P}(b = 1 \mid x, a) \propto \exp(\eta \cdot r(x, a, b = 1)) \propto \exp(\eta \cdot (R(x, y_1) - R(x, y_2)))$$
$$\mathbb{P}(b = 0 \mid x, a) \propto \exp(\eta \cdot r(x, a, b = 0)) \propto \exp(\eta \cdot (R(x, y_2) - R(x, y_1))),$$

which is exactly the Bradley-Terry model (Bradley & Terry, 1952) in the previous RLHF literature (Rafailov et al., 2024; Liu et al., 2024b; Xiong et al., 2024; Cen et al., 2024).

The objective of the leader is to maximize the human's reward with a KL regularization:

$$\max_\pi J(u, \pi) := \mathbb{E}_{x \sim \rho, a \sim \pi(\cdot | x)}[u(s, a)] - \beta \mathbb{D}_{\text{KL}}[\pi \parallel \pi_{\text{ref}}],$$

where $\pi_{\text{ref}}$ is the reference policy that usually trains with supervised fine-tuning, and the parameter $\beta$ controls the deviation between the output policy $\pi$ and the reference policy.

We parameterize the reward function $R(x, y)$ using a function class $\{R^\theta(x, y)\}_{\theta \in \Theta}$. Note that the preference feedback is only dependent on the difference $R(x, y_1) - R(x, y_2)$, hence the reward $R^*$ is only identifiable up to a global shift. Hence, we can construct a base policy $\pi_{\text{base}}$ and consider the following reward function class

$$\{R_\theta : \mathbb{E}_{x \sim \rho, a \sim \pi_{\text{base}}}[R_\theta(x, y)] = 0\}$$

Now we apply our PES algorithm to the RLHF setting. The pseudo-code is shown in Algorithm 2. The corresponding reward function of the follower and the leader are denoted as $u^\theta(x, a)$ and $r^\theta(x, a, b)$. We also denote the ground-truth reward function and the optimal policy as $R^*, u^*, r^*$ and $\pi^*$ respectively. Now in each episode $t \in [T]$, the agent first estimates the reward function $\theta^t$ using the historical data $\{D^s\}_{s \in [t-1]} = \{x^s, a^s, b^s, \pi^s\}_{s \in [t-1]}$ by maximizing $\max_\pi J(u^{\theta^t}, \pi) - \eta_1 L^t(\theta)$, where

$$L^t(\theta) = -\sum_{i=1}^{t-1} \left[ b^i \log(\sigma(\eta \cdot r^\theta(x^i, a^i, b^i))) \right]$$

is the cross-entropy loss, and $\sigma(z) = 1/(1 + \exp(-z))$ is the sigmoid function. Then the agent predicts $\pi^t$ via the optimal policy associated with $\theta^t$, and executes $\pi^t$ to collect data $D_t$. The regret $\text{Reg}(T)$ then can be defined as Equation 3.3.

---

**Algorithm 2** Planning after Estimation-RLHF (PES-RLHF)

---

1: **Initial**: $\mathcal{D} = \emptyset$.
2: **for** $t = 1, 2, \cdots, T$ **do**
3:     Calculate $\theta^t = \arg\max_\theta \left( \max_\pi J(u^{\theta^t}, \pi) - \eta_1 L^t(\theta) \right)$.
4:     Execute $\pi^t = \arg\max_{\pi \in [\Delta(\mathcal{A})]^H} J(u^{\theta^t}, \pi)$.
5:     Collect data $D_t$ with $D_t = (s^t, a^t, b^t, \pi^t)$, and update $\mathcal{D} = \mathcal{D} \cup D_t$.
6: **end for**

---

Now we can get the following theoretical result for the RLHF setting.

**Theorem 2.** *If we choose $\eta_1 = 1/\sqrt{T}$, then with probability at least $1 - \delta$, the Algorithm 2 achieves a regret*

$$\text{Reg}(T) \le 2\sqrt{T} \log \frac{|\mathcal{R}|}{\delta} + 2 \cdot (3 + e^2)^2 \eta^{-2} d\kappa \exp(2/\beta) \sqrt{T}, \tag{6.1}$$

*where $\mathcal{R}$ is the reward hypothesis function class, $\kappa = \sup_{x,y} \frac{\pi_{\text{base}}(y|x)}{\pi_{\text{ref}}(y|x)}$, and $d$ is the multi-agent decoupling coefficient in Definition 1 with*

$$\mathcal{F} = \{f : f(x, (y_1, y_2), b) = (R(x, y_1) - R(x, y_2)) - (R^*(x, y_1) - R^*(x, y_2))\}$$
$$\varrho = \{\rho \in \mathbb{P}^\pi(a_h = \cdot, x \sim \rho), \forall \pi \in \Pi\}.$$

The result above shows that the PES algorithm framework can handle the RLHF setting as a special case, and the resulting PES-RLHF algorithm is similar to the online version of RPO (Liu et al., 2024b), and the reward-based version of VPO (Cen et al., 2024). Moreover, compared to Cen et al. (2024), we only relies on the decoupling coefficient of the reward function class, rather than the stronger linear assumption. Compared to Liu et al. (2024b), they study offline setting with the coverage assumption and pessimism principle, so the first exploration term $\max_\pi J(u^{\theta^t}, \pi)$ changes the sign (Liu et al., 2024b).

## 7 Conclusion

In this paper, we propose an easy-to-implement RL algorithm, Planning after Estimation (PES) to efficiently solve MSG in the context of general function approximation. Compared to the other concurrent works , our algorithm circumvents intractable optimistic planning, which involves joint planning of the leader's policy, value function, and the follower's quantal response model. In the theoretical analysis, we prove that with a set of simple hyper-parameter choices, PES achieves a sub-linear $\tilde{O}(d_c \sqrt{T})$-regret with general function approximation, where $d_c$ is the decoupling coefficient and $T$ is the number of episodes. At last, we apply PES to the RLHF setting by formulating the RLHF as a turn-based Stackelberg game to demonstrate the efficacy of our algorithm..

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

## A  Technical Lemmas

**Lemma 1.** *Let $Z_t$ be a sequence of random variables, where each $Z_t$ may depend on the previous observations $S_{t-1} = [Z_1, ..., Z_{t-1}] \in \mathcal{Z}^{t-1}$. Furthermore, we define a filtration $\{\mathcal{F}_t = \sigma(S_t)\}$, which is also the natural filtration of $\{Z_t\}$. Consider a sequence of real-valued random (measurable) functions $\xi_1(S_1), ..., \xi_T(S_T)$. Let $\tau \leq T$ be a stopping time so that $\mathbb{I}(t \leq \tau)$ is measurable in $S_t$. We have*

$$\mathbb{E}_{S_T} \exp\left(\sum_{t=1}^{\tau} \xi_i - \sum_{t=1}^{\tau} \ln \mathbb{E}_{Z_t|S_{t-1}} e^{\xi_t}\right) = 1.$$

*Proof.* This proof is a revised version of Lemma 13.1 in Zhang (2023). We prove this lemma by induction. When $T = 0$, the equality apparently holds. We then assume that the claim holds at $T - 1$ for some $T \geq 1$. Now we will prove the equation at time $T$ using the induction hypothesis.

First we define $\tilde{\xi}_t = \xi_t \mathbb{I}(t \leq \tau)$ and notice that $\tilde{\xi}_t$ is measurable in $S_t$ so we have

$$\mathbb{E}_{S_T} \exp\left(\sum_{t=1}^{\tau} \xi_t - \sum_{i=1}^{\tau} \ln \mathbb{E}_{Z_t|S_{t-1}} e^{\xi_t}\right)$$

$$= \mathbb{E}_{S_T} \exp\left(\sum_{t=1}^{T} \tilde{\xi}_t - \sum_{i=1}^{T} \ln \mathbb{E}_{Z_t|S_{t-1}} e^{\tilde{\xi}_t}\right)$$

$$= \mathbb{E}_{S_{T-1}} \left[\exp\left(\sum_{t=1}^{T-1} \tilde{\xi}_t - \sum_{i=1}^{T-1} \ln \mathbb{E}_{Z_t|S_{t-1}} e^{\tilde{\xi}_t}\right) \mathbb{E}_{Z_T|S_{T-1}} \exp\left(\tilde{\xi}_T - \ln \mathbb{E}_{Z_t|S_{t-1}} e^{\tilde{\xi}_T}\right)\right]$$

$$= \mathbb{E}_{S_{T-1}} \left[\exp\left(\sum_{t=1}^{T-1} \tilde{\xi}_t - \sum_{i=1}^{T-1} \ln \mathbb{E}_{Z_t|S_{t-1}} e^{\tilde{\xi}_t}\right)\right]$$

$$= \mathbb{E}_{S_{T-1}} \left[\exp\left(\sum_{t=1}^{\min(\tau, T-1)} \xi_t - \sum_{i=1}^{\min(\tau, T-1)} \ln \mathbb{E}_{Z_t|S_{t-1}} e^{\xi_t}\right)\right]$$

$$= 1,$$

where the third equality exploits the fact that $\mathbb{E}_{Z_T^{(y)}} \exp\left(\tilde{\xi}_T - \ln \mathbb{E}_{Z_t|S_{t-1}} e^{\tilde{\xi}_T}\right) = 1$; and the last equality is because we could treat $\min(\tau, T-1)$ as a stopping time no more than $T - 1$ and we could use the induction hypothesis. $\square$

**Lemma 2** (Martingale exponential inequality). *For a sequence of real-valued random variables $\{X_t\}_{t \leq T}$ adapted to a filtration $\{\mathcal{F}_t\}_{t \leq T}$, the following holds with probability at least $1 - \delta$, for $\forall t \in [T]$,*

$$-\sum_{s=1}^{t} X_s \leq \sum_{s=1}^{t} \ln \mathbb{E}\left[e^{-X_s}|\mathcal{F}_{s-1}\right] + \ln \frac{1}{\delta}.$$

*And also*

$$\sum_{s=1}^{t} X_s \leq \sum_{s=1}^{t} \ln \mathbb{E}\left[e^{X_s}|\mathcal{F}_{s-1}\right] + \ln \frac{1}{\delta}.$$

*Proof.* It only suffices to show the case when $\{\xi_i\}_{i=1}^{T}$ is a finite case. The statement implies the original lemma by pushing $T \to +\infty$. Let

$$U_\tau = -\sum_{s=1}^{\tau} X_s - \sum_{s=1}^{\tau} \ln \mathbb{E}_{S_t} e^{-X_s},$$

where $\tau$ is some stopping time. By Lemma 1 we have $\mathbb{E}(\exp^{U_\tau}) = 1$. (In this case, we apply $Z_s = \xi_s = -X_s$ in Lemma 1). Now we define the stopping time $\tau$ as

$$\tau = \min\left(T, \min\left(n : U_n \geq -\ln \delta\right)\right).$$

Then it follows that

$$\mathbb{P}\left(\exists n : U_\tau \geq -\ln \delta\right) \leq \mathbb{E}\left[e^{U_\tau + \ln \delta}\right] = \delta \mathbb{E}\left[e^{U_\tau}\right] = \delta,$$

where the first inequality is by the famous Markov Inequality.

By considering the complementary event, we know with probability at least $1 - \delta$, the following inequality holds for any $t \in [T]$

$$-\sum_{s=1}^{t} X_s \leq \sum_{s=1}^{t} \ln \mathbb{E}\left[e^{-X_s} | \mathcal{F}_{s-1}\right] + \ln \frac{1}{\delta}.$$

$\square$

**Lemma 3** (Freedman's inequality). *Let $\{X_t\}_{t \leq T}$ be any sequence of real-valued random variables adapted to filtration $\{\mathcal{F}_t\}_{t \leq T}$. If $|X_t| \leq R$ almost surely, then for any $\eta \in (0, \frac{1}{2R}]$ it holds that with probability at least $1 - \delta$,*

$$\sum_{t=1}^{T} X_t \leq \sum_{t=1}^{T} \mathbb{E}(X_t | \mathcal{F}_{t-1}) + \eta \sum_{t=1}^{T} \mathrm{Var}\left[X_t | \mathcal{F}_{t-1}\right] + \frac{\ln \frac{1}{\delta}}{\eta}.$$

*Furtheremore, we have*

$$\sum_{t=1}^{T} \mathbb{E}(X_t | \mathcal{F}_{t-1}) \leq \sum_{s=1}^{T} X_s + \eta \sum_{s=1}^{T} \mathrm{Var}\left[X_s | \mathcal{F}_{s-1}\right] + \frac{\ln \frac{1}{\delta}}{\eta}.$$

*Proof.* For any random variable $X$ we assume $|X| \leq R$ almost surely, and let $X' = X - \mathbb{E}X$. We then get $|X'| \leq 2R$ almost surely, and we have

$$\begin{aligned}
\ln \mathbb{E}\left[e^{\lambda X}\right] &= \lambda \mathbb{E}X + \ln \mathbb{E}e^{\lambda X'} \\
&\leq \lambda \mathbb{E}X + \mathbb{E}e^{\lambda X'} - 1 \\
&= \lambda \mathbb{E}X + \lambda^2 \mathbb{E}\left[\frac{e^{\lambda X' - \lambda X' - 1}}{(\lambda X')^2}(X')^2\right] \\
&\leq \lambda \mathbb{E}X + \lambda^2 \phi(\lambda 2R) \mathrm{Var}\left[X\right],
\end{aligned}$$

where $\phi(x) = \frac{e^x - x - 1}{x^2}$; the first inequality uses $\ln x \leq x - 1$; the second inequality exploits the fact that $\phi(x)$ is non-decreasing. Then, we consider the Taylor expansion: $e^x = \sum_{n=0}^{\infty} \frac{x^n}{n!}$, and we have

$$\phi(x) = \sum_{n=2}^{\infty} \left(\frac{x^{n-2}}{n!}\right) \leq \frac{1}{2} \sum_{n=0}^{\infty} \left(\frac{x}{2}\right)^n.$$

For any $\lambda \in (0, \frac{1}{2R}]$, we could get a finite upper bound for $\ln \mathbb{E}\left[e^{\lambda X}\right]$:

$$\ln \mathbb{E}\left[e^{\lambda X}\right] \leq \lambda \mathbb{E}X + \lambda^2 \frac{1}{2} \sum_{n=0}^{\infty} (\lambda R)^n \mathrm{Var}\left[X\right] = \lambda \mathbb{E}X + \frac{\lambda^2 \mathrm{Var}\left[X\right]}{2(1 - \lambda R)}. \tag{A.1}$$

Similar to Lemma 2, we let

$$V_\tau(\lambda) = \lambda \sum_{s=1}^{\tau} X_s - \sum_{s=1}^{\tau} \ln \mathbb{E}_{S_t} e^{\lambda X_s},$$

where $\tau$ is some stopping time. By Lemma 1 we have $\mathbb{E}(\exp^{V_\tau(\lambda)}) = 1$. (In this case, we apply $Z_s = \xi_s = X_s$ in Lemma 1). Now we define the stopping time $\tau$ as

$$\tau = \min\left(T, \min\left(n : V_n(\lambda) \geq -\ln \delta\right)\right).$$

Then it follows that

$$\mathbb{P}\left(\exists n : V_\tau(\lambda) \geq -\ln \delta\right) \leq \mathbb{E}\left[e^{V_\tau(\lambda) + \ln \delta}\right] = \delta \mathbb{E}\left[V_\tau(\lambda)\right] = \delta,$$

where the first inequality is by the famous Markov Inequality.

By considering the complementary event, we know with probability at least $1 - \delta$, the following inequality holds

$$\sum_{s=1}^{T} X_s \leq \frac{1}{\lambda} \left( \sum_{s=1}^{T} \ln \mathbb{E}\left[ e^{\lambda X_s} | \mathcal{F}_{s-1} \right] + \ln \frac{1}{\delta} \right).$$

Then we take $\lambda = \eta \in (0, \frac{1}{2R}]$ and use equation (A.1) to prove the original statement:

$$\sum_{s=1}^{T} X_s \leq \sum_{t=1}^{T} \mathbb{E}(X_t | \mathcal{F}_{t-1}) + \frac{\eta \sum_{s=1}^{T} \mathrm{Var}\left[ X_s | \mathcal{F}_{s-1} \right]}{2(1 - \eta R)} + \frac{\ln \frac{1}{\delta}}{\eta}$$

$$\leq \sum_{t=1}^{T} \mathbb{E}(X_t | \mathcal{F}_{t-1}) + \eta \sum_{s=1}^{T} \mathrm{Var}\left[ X_s | \mathcal{F}_{s-1} \right] + \frac{\ln \frac{1}{\delta}}{\eta}.$$

By letting $X'_s = -X_s$, we could easily get

$$\sum_{t=1}^{T} \mathbb{E}(X_t | \mathcal{F}_{t-1}) \leq \sum_{s=1}^{T} X_s + \eta \sum_{s=1}^{T} \mathrm{Var}\left[ X_s | \mathcal{F}_{s-1} \right] + \frac{\ln \frac{1}{\delta}}{\eta}.$$

$\square$

**Lemma 4** (Elliptical Potential Lemma). *Let $\{x_s\}_{s \in [k]}$ be a sequence of vectors with $x_s \in \mathcal{V}$ for some Hilbert space $\mathcal{V}$. Let $\Lambda_0$ be a positive definite matrix and define $\Lambda_k = \Lambda_0 + \sum_{s=1}^{k} x_s x_s^\top$. Then it holds that*

$$\sum_{s=1}^{k} \min \left\{ 1, \|x_s\|_{\Lambda_s^{-1}} \right\}^2 \leq 2 \ln \left( \frac{\det(\Lambda_{K+1})}{\det(\Lambda_0)} \right).$$

*Proof.* This proof mainly follows Lemma 11 in Abbasi-Yadkori et al. (2011). By simple calculation, we have

$$\det(\Lambda_k) = \det(\Lambda_{k-1} + x_k x_k^\top) = \det(\Lambda_{k-1}) \det(I + \Lambda_{k-1}^{-\frac{1}{2}} x_k (\Lambda_{k-1}^{-\frac{1}{2}} x_k)^\top)$$

$$= \det(\Lambda_{k-1})(1 + \|x_{n-1}\|_{\Lambda_{k-1}^{-1}}^2) = \det(\Lambda_0) \prod_{s=1}^{k} \left( 1 + \|s_s\|_{\Lambda_{s-1}^{-1}}^2 \right),$$

where we use the fact that all eigenvalues of a matrix of the form $I + x x^\top$ are 1 except one eigenvalue, which is $1 + \|x\|_2^2$ and which corresponds to the eigenvector $x$. Using $\log(1 + t) \leq t$, we can bound $\log(\det(\Lambda_k))$ by

$$\log \det(\Lambda_k) \leq \log \det(\Lambda_0) + \sum_{s=1}^{k} \|x_s\|_{\Lambda_{s-1}^{-1}}^2.$$

Combining $x \leq 2 \log(1 + x)$ when $x \in [0, 1]$, we get

$$\sum_{s=1}^{k} \min \left( 1, \|x_s\|_{\Lambda_{s-1}^{-1}}^2 \right) \leq 2 \sum_{t=1}^{n} \log \left( 1 + \|x_s\|_{\Lambda_{s-1}^{-1}}^2 \right) = 2 \ln \left( \frac{\det(\Lambda_k)}{\det(\Lambda_0)} \right).$$

$\square$

**Lemma 5.** *(Lemma G.2 of Chen et al. (2023)) We consider a fixed policy $\pi$ and Let $\tilde{Q}$ be an estimate of $Q^\pi$. We define a V-function $\tilde{V}$ and an advantage function $\tilde{A}$ by letting*

$$\tilde{V}_h(s) = \frac{1}{\eta} \log \left( \sum_{b \in \mathcal{B}} \exp(\eta \cdot \tilde{Q}_h(s, b)) \right), \qquad \tilde{A}_h(s, a) = \tilde{Q}_h(s, b) - \tilde{V}_h(s).$$

*Furthermore, we define a follower's policy $\tilde{\upsilon}$ be letting $\tilde{\upsilon}_h(b|s) = \exp(\eta \cdot \tilde{A}_h(s, b))$. Then we have*

$$\mathrm{D_H}\left( \upsilon^\pi, \tilde{\upsilon} \right) \geq \frac{\eta^2}{8(1 + \eta B_A)^2} \cdot \left\langle \upsilon^\pi, (\tilde{A} - A)^2 \right\rangle_{\mathcal{B}}.$$

*where $B_A = 2 \left( \eta^{-1} \log |\mathcal{B}| + 1 \right)$.*

**Lemma 6.** *For any $h \in [H]$ and $(s_h, b_h) \in \mathcal{S} \times \mathcal{B}$, using the same notation as in Lemma 5, we have*

$$A_h^\pi(s_h, b_h) - \tilde{A}_h(s_h, b_h) = (\mathbb{E}_{s_h, b_h} - \mathbb{E}_{s_h})\left[Q_h(s_h, b_h)^\pi - \tilde{Q}_h(s_h, b_h)\right] + \frac{1}{\eta}\mathrm{KL}\left(\upsilon_h^\pi \| \tilde{\upsilon}_h\right).$$

*Proof.* This proof mainly follows Lemma G.4 in Chen et al. (2023). At first, we notice the fact that

$$\frac{1}{\eta}\mathcal{H}(\upsilon_h^\pi) = -\frac{1}{\eta}\left\langle \upsilon_h^\pi, \log \upsilon_h^\pi \right\rangle_{\mathcal{B}} = -\left\langle \upsilon_h^\pi, Q_h^\pi(s_h, b_h) - V_h^\pi(s_h) \right\rangle_{\mathcal{B}}, \tag{A.2}$$

$$\frac{1}{\eta}\mathcal{H}(\tilde{\upsilon}_h) = -\frac{1}{\eta}\left\langle \tilde{\upsilon}_h^\pi, \log \tilde{\upsilon}_h \right\rangle_{\mathcal{B}} = -\left\langle \tilde{\upsilon}_h, \tilde{Q}_h(s_h, b_h) - \tilde{V}_h(s_h) \right\rangle_{\mathcal{B}}. \tag{A.3}$$

Then we could write the difference of V-functions as

$$V_h^\pi(s_h) - \tilde{V}_h(s_h)$$
$$= \left\langle \upsilon_h^\pi, V_h^\pi(s_h) \right\rangle_{\mathcal{B}} - \left\langle \tilde{\upsilon}_h, \tilde{V}_h(s_h) \right\rangle_{\mathcal{B}}$$
$$= \left\langle \upsilon_h^\pi, Q_h^\pi(s_h, b_h) \right\rangle_{\mathcal{B}} + \frac{1}{\eta}\mathcal{H}(\upsilon_h^\pi) - \left\langle \tilde{\upsilon}_h, \tilde{Q}_h(s_h, b_h) \right\rangle_{\mathcal{B}} - \frac{1}{\eta}\mathcal{H}(\tilde{\upsilon}_h)$$
$$= \left\langle \upsilon_h^\pi, Q_h^\pi(s_h, b_h) - \tilde{Q}_h(s_h, b_h) \right\rangle_{\mathcal{B}} + \left\langle \upsilon_h^\pi - \tilde{\upsilon}_h, \tilde{Q}_h(s_h, b_h) \right\rangle_{\mathcal{B}}$$
$$\quad - \left\langle \upsilon_h^\pi, Q_h^\pi(s_h, b_h) - V_h^\pi(s_h) \right\rangle_{\mathcal{B}} + \left\langle \tilde{\upsilon}_h, \tilde{Q}_h(s_h, b_h) - \tilde{V}_h(s_h) \right\rangle_{\mathcal{B}},$$

where the first equality exploits the fact that $V_h(s_h)$ is constant w.r.t. $b_h \in \mathcal{B}$ and $\upsilon_h^\pi, \tilde{\upsilon}_h$ are probability distributions on $\mathcal{B}$; the second equality is by equation (A.2); the last equality is by simple algebraic tricks.

Then, by direct calculation and omitting $(s_h, b_h)$ for $Q_h^\pi, \tilde{Q}_h$ and $(s_h)$ for $V_h, \tilde{V}_h$, we have

$$-\left\langle \upsilon_h^\pi, Q_h^\pi - V_h^\pi - (\tilde{Q}_h - \tilde{V}_h) \right\rangle_{\mathcal{B}} = \left\langle \upsilon_h^\pi - \tilde{\upsilon}_h, \tilde{Q}_h \right\rangle_{\mathcal{B}} - \left\langle \upsilon_h^\pi, Q_h^\pi - V_h^\pi \right\rangle_{\mathcal{B}} + \left\langle \tilde{\upsilon}_h, \tilde{Q}_h - \tilde{V}_h \right\rangle_{\mathcal{B}},$$

where we use the fact $\left\langle \upsilon_h^\pi, \tilde{V}_h \right\rangle_{\mathcal{B}} = \left\langle \tilde{\upsilon}_h, \tilde{V}_h \right\rangle_{\mathcal{B}}$, since $\tilde{V}_h$ is a constant w.r.t. $b_h \in \mathcal{B}$. Therefore, we can write $V_h^\pi(s_h) - \tilde{V}_h(s_h)$ as

$$V_h^\pi(s_h) - \tilde{V}_h(s_h)$$
$$= \left\langle \upsilon_h^\pi, Q_h^\pi(s_h, b_h) - \tilde{Q}_h(s_h, b_h) \right\rangle_{\mathcal{B}} - \left\langle \upsilon_h, Q_h^\pi(s_h, b_h) - V_h^\pi(s_h) - (\tilde{Q}_h(s_h, b_h) - \tilde{V}_h(s_h)) \right\rangle_{\mathcal{B}}$$
$$= \left\langle \upsilon_h^\pi, Q_h^\pi(s_h, b_h) - \tilde{Q}_h(s_h, b_h) \right\rangle_{\mathcal{B}} - \left\langle \upsilon_h, A_h^\pi(s_h, b_h) - \tilde{A}_h(s_h, b_h) \right\rangle_{\mathcal{B}}$$
$$= \left\langle \upsilon_h^\pi, Q_h^\pi(s_h, b_h) - \tilde{Q}_h(s_h, b_h) \right\rangle_{\mathcal{B}} - \frac{1}{\eta}\mathrm{KL}\left(\upsilon_h^\pi \| \tilde{\upsilon}_h\right)_{\mathcal{B}}.$$

We notice the fact that $\mathrm{KL}\left(\upsilon_h^\pi \| \tilde{\upsilon}_h\right) = \eta \left\langle \upsilon_h^\pi, A_h^\pi(s_h, b_h) - \tilde{A}_h(s_h, b_h) \right\rangle_{b \in \mathcal{B}}$. At last, we could get

$$A_h^\pi(s_h, b_h) - \tilde{A}_h(s_h, b_h) = (\mathbb{E}_{s_h, b_h} - \mathbb{E}_{s_h})\left[Q_h^\pi(s_h, b_h) - \tilde{Q}_h(s_h, b_h)\right] + \frac{1}{\eta}\mathrm{KL}\left(\upsilon_h^\pi \| \tilde{\upsilon}_h\right).$$

$\square$

**Lemma 7.** *We define a distance $\rho_1$ on $\Theta$ by letting*

$$\rho_1(\theta, \tilde{\theta}) := \max_{\pi \in \Pi, s_h \in \mathcal{S}, h \in [H]}\left\{D_\mathrm{H}\left(\upsilon_h^{\pi, \theta}(\cdot | s_h), \upsilon_h^{\pi, \tilde{\theta}}(\cdot | s_h)\right), (1 + \eta)\left\| r_h^{\pi, \theta} - r_h^{\pi, \tilde{\theta}} \right\|_\infty\right\}. \tag{A.4}$$

*Let $\mathcal{N}_{\rho_1}(\theta, \epsilon)$ be the $\epsilon$-covering number of $\Theta$ with respect to the distance $\rho_1$. For any $\delta \in (0, 1)$, we set $\beta_1 = 2\ln(H \cdot \mathcal{N}(\Theta, T^{-1})/\delta) + 8$. For $\forall \theta \in \Theta, \forall h \in [H]$,*

$$\sum_{i=1}^{t-1}\mathbb{E}^{\pi^i}\mathrm{Var}_{s_h}^{\pi^i, \theta^\star}\left[r_h^{\pi^i, \theta}(s_h, b_h) - r_h^{\pi^i, \theta^\star}(s_h, b_h)\right] \leq 4C_\eta^2\left(L_{h,1}^t(\theta) - L_{h,1}^t(\theta^\star)\right) + \beta,$$

*where we define*

$$\mathrm{Var}_{s_h}^{\pi, \theta}[Z] = \mathrm{Var}^{\pi, \theta}[Z | s_h] = \mathbb{E}^{\pi_h, \upsilon_h^{\pi, \theta}}\left[(Z - \mathbb{E}^{\pi_h, \upsilon_h^{\pi, \theta}}[Z | s_h])^2 | s_h\right],$$

$$C_\eta = \frac{1}{\eta} + B_A, B_A = 2(\eta^{-1}\log|\mathcal{B}| + 1).$$

*Proof.* We first exploit Lemma 2 with $X_t^h = \frac{1}{2}(\log \upsilon_h^{\pi_i,\theta}(s_h^t|b_h^t) - \log \upsilon_h^{\pi_i,\theta^\star}(s_h^t|b_h^t))$. We choose filtration to be $\mathcal{F}_{h;t-1}\{X_i^h : i \in [t-1]\}$. Let $\mathcal{N}_{\rho_1}(\Theta, \epsilon)$ be the covering number for the $\epsilon$-covering net of $\Theta$ with respect to norm $\rho_1$ defined in A.4. Let $\Theta_\epsilon$ be the $\epsilon$-covering net of $\Theta$. By Lemma 2, w.p. at least $1 - \delta$, for a fixed $\theta \in \Theta_\epsilon$ and a fixed $h \in [H]$, we have

$$\sum_{t=1}^{t-1} X_t^h = \frac{1}{2}(L_{h,1}^t(\theta^\star) - L_{h,1}^t(\theta))$$

$$\overset{(a)}{\leq} \sum_{t=1}^{t-1} \log \mathbb{E}(e^{X_t}|\mathcal{F}_{t-1}) + \frac{1}{\delta}$$

$$\overset{(b)}{=} \sum_{i=1}^{t-1} \log \mathbb{E}^{\pi^i}\left[\sqrt{\frac{\upsilon^{\pi_i,\theta}(\cdot|s_h)}{\upsilon^{\pi_i,\theta^\star}(\cdot|s_h)}}\right] + \frac{1}{\delta}$$

$$\overset{(c)}{\leq} -\sum_{i=1}^{t-1} \mathbb{E}^{\pi^i}\left[D_H^2\left(\upsilon^{\pi_i,\theta}(\cdot|s_h), \upsilon^{\pi_i,\theta^\star}(\cdot|s_h)\right)\right] + \frac{1}{\delta},$$

where the first equality is by the definition of $L_{h,1}^t$; (a) is by Lemma 2; (b) is by the definition of $X_t$; (c) is by the fact that $\log(x) \leq x - 1$ and the definition of Hellinger distance.

By taking union bound on $\theta \in \Theta_\epsilon$ and $h \in [H]$, we have for any $\theta \in \Theta$, any $h \in [H]$, with probability at least $1 - \delta$, for $\forall t \in [T]$

$$\frac{1}{2}(L_{h,1}^t(\theta^\star) - L_{h,1}^t(\theta)) \leq -\sum_{i=1}^{t-1} \mathbb{E}^{\pi^i}\left[D_H^2\left(\upsilon^{\pi_i,\theta}(\cdot|s_h), \upsilon^{\pi_i,\theta^\star}(\cdot|s_h)\right)\right] + \frac{\log\left(H\mathcal{N}_\rho(\Theta, \epsilon)\right)}{\delta}. \quad (A.5)$$

On the other hand, by the definition of $\rho_1$ in equation (A.4), for any $\theta, \tilde{\theta} \in \Theta$, we have

$$\left|D_H^2\left(\upsilon_h^{\pi,\theta}, \upsilon_h^{\pi,\theta^\star}\right) - D_H^2\left(\upsilon_h^{\pi,\tilde{\theta}}, \upsilon_h^{\pi,\theta^\star}\right)\right|$$

$$\overset{(a)}{=} \left|D_H\left(\upsilon_h^{\pi,\theta}, \upsilon_h^{\pi,\theta^\star}\right) + D_H\left(\upsilon_h^{\pi,\tilde{\theta}}, \upsilon_h^{\pi,\theta^\star}\right)\right| \cdot \left|D_H\left(\upsilon_h^{\pi,\theta}, \upsilon_h^{\pi,\theta^\star}\right) - D_H\left(\upsilon_h^{\pi,\tilde{\theta}}, \upsilon_h^{\pi,\theta^\star}\right)\right|$$

$$\overset{(b)}{\leq} 2D_H\left(\upsilon_h^{\pi,\tilde{\theta}}, \upsilon_h^{\pi,\theta}\right)$$

$$\overset{(c)}{\leq} 2\rho_1(\theta, \tilde{\theta}),$$

where (a) is by the fact that $a^2 - b^2 = (a+b)(a-b) \leq |a+b||a-b|$; (b) is by the fact that $D_H(\cdot, \cdot) \leq 1$; (c) is by the definition of $\rho_1$. Then noting that $L_{h,1}^t(\theta) = -\sum_{i=1}^{t} \eta A_h^{\pi^i,\theta}(s_h^i, b_h^i)$, for any $\theta, \tilde{\theta} \in \Theta$, we have

$$\left|L_{h,1}^t(\theta) - L_{h,1}^t(\tilde{\theta})\right| \leq \eta T \max_{i \in [t-1]} \left|A_h^{\pi^i,\theta}(s_h^i, b_h^i) - A_h^{\pi^i,\tilde{\theta}}(s_h^i, b_h^i)\right|$$

$$\leq 2\eta T \max_{i \in [t-1]} \left\|r_h^{\pi^i,\theta} - r_h^{\pi^i,\tilde{\theta}}\right\|_\infty$$

$$\leq 2T \cdot \rho_1(\theta, \tilde{\theta}),$$

where the second inequality uses the fact that $\left|\left(V_h^{\pi,\theta} - V_h^{\pi,\tilde{\theta}}\right)(s_h)\right| \leq \left\|r_h^{\pi,\theta} - r_h^{\pi,\tilde{\theta}}\right\|_\infty$; and the last inequality is by the definition of $\rho_1$. Therefore, all the error terms in $D_H^2(\cdot, \cdot)$, $L_{h,1}^t(\theta^\star)$ and $L_{h,1}^t(\theta)$ induced by $\epsilon$-net could be bounded by $2T\epsilon$. By adding an extra $4T\epsilon$ in equation (A.5), we have for all $\theta \in \Theta, h \in [H], t \in [T]$, w.p. $1 - \delta$,

$$\frac{1}{2}(L_{h,1}^t(\theta^\star) - L_{h,1}^t(\theta)) \leq -\sum_{i=1}^{t-1} \mathbb{E}^{\pi^i}\left[D_H^2\left(\upsilon^{\pi_i,\theta}(\cdot|s_h), \upsilon^{\pi_i,\theta^\star}(\cdot|s_h)\right)\right] + \frac{\log\left(H\mathcal{N}_\rho(\Theta, \epsilon)\right)}{\delta} + 4T\epsilon.$$

$$(A.6)$$

In the rest of the proof we take $\epsilon = \frac{1}{T}$ and let $\beta_1 = 2\log(H\mathcal{N}_\rho(\Theta, T^{-1})/\delta) + 8$. By Lemma 5, we have

$$
\begin{aligned}
8\mathrm{D}_{\mathrm{H}}^2\left(v^{\pi_i,\theta}(\cdot|s_h), v^{\pi_i,\theta^\star}(\cdot|s_h)\right) &\geq \left(\frac{\eta}{1+\eta B_A}\right)^2 \cdot \left\langle v_h^{\pi,\theta^\star}, (A_h^{\pi,\theta} - A_h^{\pi,\theta^\star})^2\right\rangle \\
&\geq \left(\frac{\eta}{1+\eta B_A}\right)^2 \cdot \mathbb{E}_{s_h}^{\pi,\theta^\star}\left(A_h^{\pi,\theta} - A_h^{\pi,\theta^\star}\right)^2 \\
&\geq \left(\frac{\eta}{1+\eta B_A}\right)^2 \cdot \mathbb{E}_{s_h}^{\pi,\theta^\star}\left((\mathbb{E}_{s_h,b_h}^{\pi,\theta^\star} - \mathbb{E}_{s_h}^{\pi,\theta^\star})[r_h^{\pi,\theta} - r_h^{\pi,\theta^\star}]\right)^2 \\
&= \left(\frac{\eta}{1+\eta B_A}\right)^2 \cdot \mathrm{Var}_{s_h}^{\pi_i,\theta^\star}[r_h^{\pi,\theta}(s_h,b_h) - r_h^{\pi,\theta^\star}(s_h,b_h)],
\end{aligned}
$$

where the second inequality is by Jensen's inequality of $x^2$; the last inequality is by Lemma 6; the last equality is by the definition of $\mathrm{Var}_{s_h}^{\pi_i,\theta}(\cdot)$. Therefore, by letting $C_\eta = \frac{1}{\eta} + B_A$ and insert the above result back to equation (A.6), we have

$$
\sum_{i=1}^{t-1}\mathbb{E}^{\pi_i}\mathrm{Var}_{s_h}^{\pi_i,\theta^\star}\left[r_h^{\pi_i,\theta}(s_h,b_h) - r_h^{\pi_i,\theta^\star}(s_h,b_h)\right] \leq 4C_\eta^2(L_{h,1}^t(\theta) - L_{h,1}^t(\theta^\star)) + \beta.
$$

$\square$

**Lemma 8.** *Let $\mathcal{F}_h = \mathcal{U}_h \times \mathcal{U}_{h+1} \times \Theta$, we define the following distance on for $f, \tilde{f} \in \mathcal{F}_h$:*

$$
\rho_2(f, \tilde{f}) := \max_{h\in[H]}\left\{\left\|U_h - \tilde{U}_h\right\|_\infty, \left\|T_{h+1}^{\star,\theta}U(h+1)(\cdot) - T_{h+1}^{\star,\tilde\theta}\tilde{U}(h+1)(\cdot)\right\|_\infty\right\}. \tag{A.7}
$$

*Let $\mathcal{N}_{\rho_2}(\theta, \epsilon)$ be the $\epsilon$-covering number of $\mathcal{F}$ with respect to the distance $\rho_2$. For any $\delta \in (0,1)$, we set $\beta_2 = 4H^2\ln(\frac{H\mathcal{N}_{\rho_2}(\mathcal{F},\epsilon)}{\delta}) + 5$. For $\forall\{f_h^t\}_{h\in[H],t\in[T]} \subseteq \mathcal{F}$*

$$
L_{h,2}^{t-1}(f_h^\star) - L_{h,2}^{t-1}(f_h^t) \leq -\frac{1}{2}\sum_{i=1}^{t-1}\mathbb{E}^{\pi_i}\left[\left(U_h - \mathbb{T}_{h+1}^{*,\theta^t}U_{h+1}\right)(s_h,a_h,b_h)^2\right] + \beta_2.
$$

*Proof.* At first we verify our loss $l_h^t$ satisfies generalized Bellman completeness and boundedness, which is defined as follows:

**Assumption 4.** *The function $l : \mathcal{U}_h \times \mathcal{U}_{h+1} \times \Theta \times (\mathcal{S}\times\mathcal{A}\times\mathcal{B}\times\mathbb{R}\times\mathcal{S}) \to \mathbb{R}$ satisfies:*

*1. (Generalized Bellman Completeness) There exists a functional operator $\mathcal{P}_h : \mathcal{H}_{h+1} \to \mathcal{H}_h$ such that for any $(U_h, U_{h+1}, \theta) \in \mathcal{H}_h \times \mathcal{H}_{h+1} \times \Theta$ and $D_h = (s_h, a_h, b_h, s_{h+1}) \in (\mathcal{S}\times\mathcal{A}\times\mathcal{B}\times\mathbb{R}\times\mathcal{S})$.*

$$
l(U_h, U_{h+1}, \theta; D_h) - l(\mathcal{P}_h U_{h+1}, U_{h+1}, \theta; D_h) = \mathbb{E}_{s_{h+1}\sim\mathbb{P}_h(\cdot|s_h,a_h,b_h)}\left[l(U_h, U_{h+1}, \theta; D_h)\right],
$$

*where we require $\mathcal{P}_h U_{h+1}^\star = U_h^\star$ and that $\mathcal{P}_h U_{h+1} \in \mathcal{H}_h$ for any $U_{h+1} \in \mathcal{U}_{h+1}$ and $h \in [H]$;*

*2. (Boundedness) It holds that $|l(U_h, U_{h+1}, \theta; D_h)| \leq B_l$ for some $B_l > 0$ and for any $(U_h, U_{h+1}, \theta) \in \mathcal{H}_h \times \mathcal{H}_{h+1} \times \Theta$ and $D_h = (s_h, a_h, b_h, s_{h+1}) \in (\mathcal{S}\times\mathcal{A}\times\mathcal{B}\times\mathbb{R}\times\mathcal{S})$.*

First we verify the Generalized Bellman Completeness:

$$
\begin{aligned}
&l_h^t(U_h, U_{h+1}, \theta; D_h^t) - \mathbb{E}_{s_{h+1}\sim\mathbb{P}_h(\cdot|s_h,a_h,b_h)}\left[l_h^t(U_h, U_{h+1}, \theta; D_h)\right] \\
&= [(U_h - u_h)(s_h, a_h, b_h) - T^{\star,\theta}(s_{h+1})] - [U_h(s_h, a_h, b_h) - \mathbb{T}_{h+1}^{\star,\theta}(U_{h+1})] \\
&= \mathbb{T}_{h+1}^{\star,\theta}(U_{h+1}) - T^{\star,\theta}(U_{h+1})(s_{h+1}) \\
&= (\mathbb{E}_{s_{h+1}\sim\mathbb{P}(\cdot|s_h,a_h,b_h)}[T^{\star,\theta}(U_{h+1})(s_{h+1})] - u_h)(s_h, a_h, b_h) - T^{\star,\theta}(U_{h+1})(s_{h+1}).
\end{aligned}
$$

Therefore, the operator $\mathcal{P}_h$ is $\mathbb{E}_{s_{h+1}\sim\mathbb{P}(\cdot|s_h,a_h,b_h)}[T^{\star,\theta}(\cdot)(s_{h+1})]$ and the generalized Bellman completeness holds. To check boundedness, we only need to notice that $u_h \in [0,1], \forall h \in [H]$, so

$|l_h^t(U_h, U_{h+1}, \theta; D_h^t)| \leq H, \forall h \in [H]$. Then we generalize the proof of Proposition 5.1 in Liu et al. (2024a) to show our wanted result.

We define the random variables $X_{h,f}^t$ as

$$X_{h,f}^t = l_h^t(U_h, U_{h+1}, \theta; D_h^t)^2 - l_h^t(\mathcal{P}_h U_{h+1}, U_{h+1}, \theta; D_h^t)^2. \tag{A.8}$$

For any $f = (U_h, U_{h+1}, \theta) \in \mathcal{U}_h \times \mathcal{U}_{h+1} \times \Theta$ and the operator $\mathcal{P}_h$ is defined as above. We first show $X_{h,f}^t$ is an unbiased estimator of the discrepancy function $d_h^t(U_h, U_{h+1}; D_h^t)^2$, which is defined as

$$d_h^t(f; D_h^t) = \mathbb{E}_{s_{h+1} \sim \mathbb{P}_h(\cdot|s_h, a_h, b_h)}[l_h^t(f; D_h^t)] = U_h - \mathbb{T}_h^{\star, \theta}(U_{h+1}).$$

For simplicity we also let $f_{\mathcal{P}} = (\mathcal{P}_h U_{h+1}, U_{h+1}, \theta)$Consider that

$$\begin{aligned}
l_h^t(f; D_h^t)^2 &= \left( l_h^t(f; D_h^t) - l_h^t(f_{\mathcal{P}}; D_h^t) + l_h^t(\mathcal{P}_h U_{h+1}, U_{h+1}, \theta; D_h^t) \right)^2 \\
&= \left( d_h^t(f; D_h^t) + l_h^t(f_{\mathcal{P}}; D_h^t) \right)^2 \\
&= (d_h^t(f; D_h^t))^2 + l_h^t(f_{\mathcal{P}}; D_h^t)^2 + 2 d_h^t(f; D_h^t) \cdot l_h^t(f_{\mathcal{P}}, \theta; D_h^t), \tag{A.9}
\end{aligned}$$

where the second equality is by the generalized Bellman completeness. Exploiting the completeness again, we have

$$\begin{aligned}
&\mathbb{E}_{s_{h+1} \sim \mathbb{P}_h(\cdot|s_h, a_h, b_h)} \left[ d_h^t(f; D_h^t) \cdot l_h^t(f_{\mathcal{P}}; D_h^t) \right] \\
&= d_h^t(f; D_h^t) \cdot \mathbb{E}_{s_{h+1} \sim \mathbb{P}_h(\cdot|s_h, a_h, b_h)} \left[ l_h^t(\mathcal{P}_h U_{h+1}, U_{h+1}, \theta; D_h^t) \right] \\
&= d_h^t(f; D_h^t) \cdot \mathbb{E}_{s_{h+1} \sim \mathbb{P}_h(\cdot|s_h, a_h, b_h)} \left[ d_h^t(f; D_h^t) - l_h^t(f; D_h^t) \right] \\
&= 0.
\end{aligned}$$

Inserting the result back to A.9, we have

$$\mathbb{E}_{s_{h+1} \sim \mathbb{P}_h(\cdot|s_h, a_h, b_h)}[X_{h,f}^t] = d_h^t(f; D_h^t)^2. \tag{A.10}$$

Then for each time step $h$, we define the filtration $\{\mathcal{F}_{h,t}\}_{t=1}^T$ with

$$\mathcal{F}_{h,t} = \sigma \left( \sum_{s=1}^k \sum_{h=1}^H D_h^s \right),$$

where $D_h^t = \{s_h^t, a_h^t, b_h^t, u_h^t, s_{h+1}^t\}$. From the previous arguments, we can derive that

$$\mathbb{E}[X_{h,f}^t \mid \mathcal{F}_{h,t-1}] = \mathbb{E} \left[ \mathbb{E}_{s_{h+1} \sim \mathbb{P}_h(\cdot|s_h, a_h, b_h)}[X_{h,f}^t] | \mathcal{F}_{h,t-1} \right] = \mathbb{E}^{\pi^t}[d_h^t(f; D_h^t)^2], \tag{A.11}$$

$$\text{Var} \left[ X_{h,f}^t \mid \mathcal{F}_{h,t-1} \right] \leq \mathbb{E}[X_{(h,f}^t)^2 \mid \mathcal{F}_{h,t-1}] \leq B_l^2 \mathbb{E}[X_{h,f}^t \mid \mathcal{F}_{h,t-1}] = B_l^2 \mathbb{E}^{\pi^t}[d_h^t(f; D_h^t)^2], \tag{A.12}$$

where $\mathbb{E}^{\pi^t}$ means the data $D_h^t$ is generated by measure $(\pi^t, \upsilon_h^{\pi^t}, P_h)$. By Lemma 3, $(|X_{h,f}^t| \leq B_l^2$ and we set $\eta = \frac{1}{2B_l^2})$, for any fixed $h \in [H], t \in [T], U \in \mathcal{U}$, we have

$$\left| \sum_{s=1}^{t-1} \mathbb{E}[X_{h,f}^s \mid \mathcal{F}_{h,t-1}] - \sum_{s=1}^{t-1} X_{h,f}^s \right| \leq \frac{1}{2B_l^2} \sum_{s=1}^{t-1} \text{Var} \left[ X_{h,f}^s \mid \mathcal{F}_{h,s-1} \right] + 2B_l^2 \log(\frac{1}{\delta})$$

$$\leq \frac{1}{2} \sum_{s=1}^{t-1} \mathbb{E}^{\pi^s}[d_h^s(f; D_h^s)^2] + 2B_l^2 \log(\frac{1}{\delta}).$$

Rearranging the above terms, we can get

$$-\sum_{s=1}^{t-1} X_{h,f}^s \leq -\frac{1}{2} \mathbb{E}^{\pi^t}[d_h^t(f; D_h^t)^2] + 2B_l^2 \log(\frac{1}{\delta}).$$

By the definition of $X_{h,f}^t$ and the loss function $L_{h,2}^t$ in (4.8), we have

$$\sum_{s=1}^{t-1} X_{h,f}^s = \sum_{s=1}^{k-1} l_h^s(f; D_h^s)^2 - \sum_{s=1}^{k-1} l_h^s(\mathcal{P}_h U_{h+1}, U_{h+1}, \theta; D_h^s)^2$$

$$\leq \sum_{s=1}^{k-1} l_h^s(f; D_h^s)^2 - \inf_{U_h' \in \mathcal{U}_h} l_h^s(U_h', U_{h+1}, \theta; D_h^s)^2$$

$$= L_{h,2}^t(f).$$

Then we can derive that, for any fixed $h \in [H], t \in [T], f \in \mathcal{U}_h \times \mathcal{U}_{h+1} \times \Theta$.

$$-L_{h,2}^t(f) \leq -\frac{1}{2}\mathbb{E}^{\pi^t}[d_h^t(f; D_h^t)^2] + 2H^2 \log(\frac{1}{\delta}). \tag{A.13}$$

Then we consider $L_{h,2}^t(f^\star)$. We first define the random variables $Y_{h,f}^t$ as

$$Y_{h,f}^t = l_h^t(U_h, U_{h+1}^\star, \theta^\star; D_h^t)^2 - l_h^t(f^\star; D_h^t)^2.$$

Similarly, we could show

$$\mathbb{E}_{s_{h+1} \sim \mathbb{P}_h(\cdot|s_h, a_h, b_h)}[Y_{h,f}^t] = (d_h^t(U_h, U_{h+1}^\star, \theta^\star; D_h^t))^2.$$

Under the filtration $\{\mathcal{F}_{h,t}\}_{t=1}^T$, we can derive that

$$\mathbb{E}[Y_{h,f}^t \mid \mathcal{F}_{h,t-1}] = \mathbb{E}^{\pi^t}[d_h^t(U_h, U_{h+1}^\star, \theta^\star; D_h^t)^2],$$

$$\mathrm{Var}\left[Y_{h,f}^t \mid \mathcal{F}_{h,t-1}\right] \leq B_l^2 \mathbb{E}^{\pi^t}[d_h^t(U_h, U_{h+1}^\star, \theta^\star; D_h^t)^2].$$

By Lemma 3, ($|Y_{h,f}^t| \leq B_l^2$ and we set $\eta = \frac{1}{2B_l^2}$), for any fixed $h \in [H], t \in [T], f \in \mathcal{F}$, we have

$$-\sum_{s=1}^{t-1} Y_{h,f}^s \leq -\frac{1}{2}\sum_{s=1}^{t-1}\mathbb{E}^{\pi^s}[d_h^s(U_h, U_{h+1}^\star, \theta^\star; D_h^t)^2] + 2B_l^2 \log(\frac{1}{\delta}) \leq 2B_l^2 \log(\frac{1}{\delta}).$$

By the definition of $Y_{h,f}^t$ and the loss function $L_{h,2}^t$ in (4.8), we have

$$-\sum_{s=1}^{t-1} Y_{h,f}^s = \sum_{s=1}^{k-1} l_h^s(f^\star; D_h^s)^2 - \sum_{s=1}^{k-1} l_h^s(U_h, U_{h+1}^\star, \theta^\star; D_h^s)^2.$$

Since such inequality holds for any $U_h \in \mathcal{U}_h$, we have

$$L_{h,2}^t(f^\star) = \sup_{U_h \in \mathcal{U}_h} (-\sum_{s=1}^{t-1} Y_{h,f}^s) \leq 2B_l^2 \log(\frac{1}{\delta}).$$

Combining the above result with (A.13), for any fixed $h \in [H], t \in [T], f \in \mathcal{F}$, we have

$$L_{h,2}^t(f^\star) - L_{h,2}^t(f) \leq -\frac{1}{2}\sum_{s=1}^{t-1}\mathbb{E}^{\pi^s}[U_h^s(s_h, a_h, b_h) - \mathbb{T}^{\star,\theta^s}(U_{h+1}^s)(s_h, a_h, b_h)] + 4H^2 \log(\frac{1}{\delta}). \tag{A.14}$$

Then we generalize this result on a $\epsilon$-net $\mathcal{F}_\epsilon$ of $\mathcal{F}$. By taking union bound over all $h \in [H], \tilde{f} \in \mathcal{F}_\epsilon$ and a $\tilde{f}^\star \in \Theta_\epsilon$ such that $\rho_2(f^\star, \tilde{f}^\star) \leq \epsilon$, with probability $1 - \delta$ we have for any $t \in [T]$

$$L_{h,2}^t(\tilde{f}^\star) - L_{h,2}^t(\tilde{f})$$

$$\leq -\frac{1}{2}\sum_{s=1}^{t-1}\mathbb{E}^{\pi^s}[\tilde{U}_h^s(s_h, a_h, b_h) - \mathbb{T}^{\star,\tilde{\theta}^s}(\tilde{U}_{h+1}^s)(s_h, a_h, b_h)] + 4H^2 \log(\frac{H\mathcal{N}_{\rho_2}(\mathcal{F}, \epsilon)}{\delta}). \tag{A.15}$$

By the definition of $\rho_2$, we know

$$\left| L_{h,2}^t(\tilde{f}^\star) - L_{h,2}^t(f^\star) \right|$$

$$= \left| \sum_{s=1}^t [(\tilde{U}_h - u_h^s)(s_h^s, a_h^s, b_h^s) - T_{h+1}^{\star,\tilde{\theta}}(s_{h+1}^s)] - [U_h(s_h^s, a_h^s, b_h^s) - T_{h+1}^{\star,\theta}(U_{h+1})] \right|$$

$$\leq \left| \sum_{s=1}^t [(\tilde{U}_h - U_h)(s_h^s, a_h^s, b_h^s) - (T_{h+1}^{\star,\tilde{\theta}} - T_{h+1}^{\star,\theta})(s_{h+1}^s)] \right|$$

$$\leq t(\left\| \tilde{U}_h - U_h \right\|_\infty + \left\| T_{h+1}^{\star,\tilde{\theta}} - T_{h+1}^{\star,\theta} \right\|_\infty)$$

$$\leq 2T\rho_2(f, \tilde{f}).$$

Similarly we could get

$$\left| L_{h,2}^t(\tilde{f}) - L_{h,2}^t(f) \right| \leq 2T\rho_2(f, \tilde{f}),$$

$$\left| [\tilde{U}_h^s - \mathbb{T}^{\star,\tilde{\theta}^s}(\tilde{U}_{h+1}^s)](s_h, a_h, b_h) - [U_h^s - \mathbb{T}^{\star,\theta^s}(U_{h+1}^s)](s_h, a_h, b_h) \right| \leq 2\rho_2 T(f, \tilde{f}).$$

Then we could generate equation (A.15) from $\mathcal{F}_\epsilon$ to $\mathcal{F}$ only paying an extra cost of $5T\epsilon$. By setting $\epsilon = 1/T$, for any $h \in [H], t \in [T], f \in \mathcal{F}$, with probability $1 - \delta$ we have

$$L_{h,2}^t(f^\star) - L_{h,2}^t$$

$$\leq -\frac{1}{2} \sum_{s=1}^{t-1} \mathbb{E}^{\pi^s} [(U_h^s - \mathbb{T}^{\star,\theta^s}(U_{h+1}^s))(s_h^s, a_h^s, b_h^s)] + 4H^2 \ln(\frac{HTN_{\rho_2}(\mathcal{F}, \epsilon)}{\delta}) + 5.$$

Let $\beta_2 = 4H^2 \ln(\frac{HTN_{\rho_2}(\mathcal{F}, \epsilon)}{\delta}) + 5$, then we are done. $\qquad \square$

**Lemma 9.** *(Lemma B.2 in (Chen et al., 2023)) For any fixed policy $\pi$ and a fixed $s_1$, let $\tilde{\upsilon}$ be an estimate of the quantal response $\upsilon^\pi$ and let $\tilde{U}$ and $\tilde{W}$ be estimates of $U^\pi$ and $W^\pi$ respectively. Based on $\tilde{U}$ and $\tilde{W}$, we can estimate $J(\pi)$ by $\tilde{W}(s_1)$. Then the error of these estimators can be bounded as follows:*

$$\tilde{W}(s_1) - J(\pi) \leq \sum_{h=1}^H \mathbb{E} \left[ \tilde{U}_h(s_h, a_h, b_h) - (\mathbb{T}_h^{\pi,\tilde{\upsilon}} \tilde{U}_{h+1}) \right] + H \sum_{h=1}^H \mathbb{E} \left[ \left\| (\upsilon_h^\pi - \tilde{\upsilon})(\cdot | s_h) \right\|_1 \right].$$

*where we define*

$$\mathbb{T}_h^{\pi,\theta} U(s_h, a_h, b_h) = u_h(s_h, a_h, b_h) + \mathbb{E}_{s_{h+1} \sim P_h(\cdot | s_h, a_h, b_h)} [(T_{h+1}^{\pi,\theta} U_{h+1})(s_{h+1})],$$

$$T_h^{\pi,\theta}(U_h)(s_h) = \langle U_h(s_h, \cdot, \cdot), \pi_h \otimes \upsilon_h^{\pi,\theta}(\cdot, \cdot | s_h) \rangle.$$

*Furthermore, by $\mathbb{T}_h^{\pi,\tilde{\upsilon}} \tilde{U}_{h+1}) \leq \mathbb{T}_h^{\star,\tilde{\upsilon}} \tilde{U}_{h+1})$, we have*

$$\tilde{W}(s_1) - J(\pi) \leq \sum_{h=1}^H \mathbb{E} \left[ \tilde{U}_h(s_h, a_h, b_h) - (\mathbb{T}_h^{\star,\tilde{\upsilon}} \tilde{U}_{h+1}) \right] + H \sum_{h=1}^H \mathbb{E} \left[ \left\| (\upsilon_h^\pi - \tilde{\upsilon})(\cdot | s_h) \right\|_1 \right].$$

**Lemma 10.** *(Lemma B.1 in (Chen et al., 2023)) We consider a fixed policy $\pi$ and let $\tilde{r}$ be an estimate of $r$. We define a V-function $\tilde{V}$ and an advantage function $\tilde{A}$ by letting*

$$\tilde{V}_h(s) = \frac{1}{\eta} \log \left( \sum_{b \in \mathcal{B}} \exp(\eta \cdot \tilde{r}_h^\pi(s, b)) \right), \qquad \tilde{A}_h(s, a) = \tilde{r}_h^\pi(s, b) - \tilde{V}_h(s).$$

*Furthermore, we define a follower's policy $\tilde{\upsilon}$ be letting $\tilde{\upsilon}_h(b|s) = \exp(\eta \cdot \tilde{A}_h(s, b))$. Then the difference between $\tilde{\upsilon}$ and $\upsilon^\pi$ can be bounded by*

$$H \sum_{h=1}^H \mathbb{E} \left[ \left\| \upsilon_h^\pi - \tilde{\upsilon}(\cdot | s_h) \right\|_1 \right]$$

$$\leq C_0 \sum_{h=1}^H \mathbb{E} \left[ \left| \mathcal{T}_h^\pi(\tilde{r}_h - r_h) \right| \right] + C_1 \sum_{h=1}^H \mathbb{E} \left[ \mathcal{T}_h^\pi(\tilde{r}_h - r_h)^2 \right],$$

*where $C_1$ is defined as*

$$C_1 = \frac{\eta^2 \exp(2\eta B_A)}{2} \left(2 + \eta B_A \cdot 2\eta B_A\right),$$

*and $\mathcal{T}_h^\pi$ has been defined in equation (5.1).*

## B  PROOF OF THEOREM 1

At first, we could decompose the regret into two terms:

$$\text{Reg}(T) = \sum_{t=1}^{T} W_1^{U^*, \theta^*}(s_1) - W_1^{\pi^t}(s_1)$$

$$\leq \underbrace{\sum_{t=1}^{T} \left( W_1^{U^*, \theta^*}(s_1) - W_1^{U^t, \theta^t}(s_1) \right)}_{I_1} + \underbrace{\sum_{t=1}^{T} \left( W_1^{U^t, \theta^t}(s_1) - W_1^{\pi^t}(s_1) \right)}_{I_2}.$$

By the definition of $U^t, \theta^t$ in algorithm 1, we have

$$W_1^{U^t, \theta^t}(s_1) - \eta_1 \sum_{h=1}^{H} L_{h,1}^t(\theta_h^t) - \eta_2 \sum_{h=1}^{H} L_{h,2}^t(U_h^t, \theta_h^t)$$

$$\geq W_1^{U^*, \theta^*}(s_1) - \eta_1 \sum_{h=1}^{H} L_{h,1}^t(\theta_h^*) - \eta_2 \sum_{h=1}^{H} L_{h,2}^t(U_h^*, \theta_h^*),$$

which implies that

$$W_1^{U^*, \theta^*}(s_1) - W_1^{U^t, \theta^t}(s_1) \leq \eta_1 \sum_{h=1}^{H}(L_{h,1}^t(\theta_h^*) - L_{h,1}^t(\theta_h^t)) + \eta_2 \sum_{h=1}^{H}(L_{h,2}^t(U_h^*, \theta_h^*) - L_{h,1}^t(U_h^t, \theta_h^t)).$$

By the lemma 7, set $\beta_1 = 2\ln \mathcal{N}_\rho(\Theta, 1/T)/\delta + 8$ with the distance $\rho$ defined in Lemma 7, and let $C_\eta = \eta^{-1} + B_A$, $B_A = 2(\eta^{-1}\log|\mathcal{B}| + 1)$, then with probability at least $1 - \delta$,

$$\sum_{h=1}^{H}(L_{h,1}^t(\theta_h^*) - L_{h,1}^t(\theta_h^t))$$

$$\leq \frac{-1}{4C_\eta^2} \sum_{h=1}^{H} \sum_{i=1}^{t} \mathbb{E}^{\pi^i} \text{Var}_{s_h}^{\pi^i, \theta^*} \left[ r_h^{\pi^i, \theta^t}(s_h, b_h) - r_h^{\pi^i, \theta^*}(s_h, b_h) \right] + H\beta_1. \tag{B.1}$$

For the variance term, we have:

$$\mathbb{E}^{\pi^i} \text{Var}_{s_h}^{\pi^i, \theta^*} \left[ r_h^{\pi^i, \theta^t}(s_h, b_h) - r_h^{\pi^i, \theta^*}(s_h, b_h) \right]$$

$$= \mathbb{E}^{\pi^i} \text{Var}^{\pi^i, \theta^*} \left[ r_h^{\pi^i, \theta^t}(s_h, b_h) - r_h^{\pi^i, \theta^*}(s_h, b_h) | s_h \right]$$

$$\overset{(a)}{=} \mathbb{E}^{\pi^i} \mathbb{E}^{\pi^i, \theta^*} \left[ \left( (r_h^{\pi^i, \theta^t} - r_h)^{\pi^i, \theta^*}(s_h, b_h) - \mathbb{E}^{\pi^i, \theta^*} \left[ (r_h^{\pi^i, \theta^t} - r_h^{\pi^i, \theta^*})(s_h, b_h) \mid s_h \right] \right)^2 | s_h \right]$$

$$\overset{(b)}{=} \mathbb{E}^{\pi^i} \left[ \left( \mathcal{T}_h^{\pi^i}(r_h^{\theta^t} - r_h^*) \right)^2 (s_h, b_h) \right],$$

where (a) follows from the definition of $\text{Var}_{s_h}^{\pi, \theta}(\cdot)$, and (b) follows from the definition of $\mathcal{T}_h^\pi(\cdot)$. Insert the last term back to equation (B.1), we have:

$$\sum_{h=1}^{H}(L_{h,1}^t(\theta_h^*) - L_{h,1}^t(\theta_h^t)) \leq \frac{-1}{4C_\eta^2} \sum_{h=1}^{H} \sum_{i=1}^{t-1} \mathbb{E}^{\pi^i} [\mathcal{T}_h^{\pi^i}(r_h^{\theta^t} - r_h^*)^2] + H\beta_1.$$

By Lemma 8, set $\beta_2 = 4H^2 \ln(\frac{H\mathcal{N}_{\rho_2}(\mathcal{F},\epsilon)}{\delta}) + 5$ with the distance $\rho_2$ defined in Lemma 8, we have

$$\sum_{h=1}^{H}(L_{h,2}^t(U_h^*, \theta_h^*) - L_{h,2}^t(U_h^t, \theta_h^t)) \leq -\frac{1}{2}\sum_{h=1}^{H}\sum_{i=1}^{t-1}\mathbb{E}^{\pi^i}\left[\left(U_h - \mathbb{T}_{h+1}^{*,\theta^t}U_{h+1}\right)(s_h, a_h, b_h)^2\right] + H\beta_2. \quad \text{(B.2)}$$

We then have

$$I_1 \leq \sum_{t=1}^{T}\left(\eta_1\sum_{h=1}^{H}(L_{h,1}^t(\theta_h^*) - L_{h,1}^t(\theta_h^t)) + \eta_2\sum_{h=1}^{H}(L_{h,2}^t(U_h^*, \theta_h^*) - L_{h,2}^t(U_h^t, \theta_h^t))\right)$$

$$\leq \eta_1 \cdot \left(-\frac{1}{4C_\eta^2}\sum_{t=1}^{T}\sum_{h=1}^{H}\sum_{i=1}^{t-1}\mathbb{E}^{\pi^i}[\mathcal{T}_h^{\pi^i}(r_h^{\theta^t} - r_h^*)^2] + HT\beta_1\right)$$

$$+ \eta_2 \cdot \left(-\frac{1}{2}\sum_{t=1}^{T}\sum_{h=1}^{H}\sum_{i=1}^{t-1}\mathbb{E}^{\pi^i}\left[\left(U_h - \mathbb{T}_{h+1}^{*,\theta^t}U_{h+1}\right)(s_h, a_h, b_h)^2\right] + HT\beta_2\right) \quad \text{(B.3)}$$

To bound $I_2$, we exploit Lemma 9 and Lemma 10,

$$I_2 \overset{(a)}{\leq} \sum_{t=1}^{T}\sum_{h=1}^{H}\mathbb{E}^{\pi^t}\left[(U_h^t)(s_h, a_h, b_h) - \mathbb{T}_{h+1}^{*,\theta^t}U_{h+1}^t(s_{h+1})\right] + H\sum_{h=1}^{H}\mathbb{E}^{\pi_t}\left[\left\|(\tilde{\upsilon}_h - \upsilon_h^\pi)(\cdot|s_h)\right\|_1\right] \quad \text{(B.4)}$$

$$\overset{(b)}{\leq} \sum_{t=1}^{T}\sum_{h=1}^{H}\mathbb{E}^{\pi^t}\left[(U_h^t)(s_h, a_h, b_h) - \mathbb{T}_{h+1}^{*,\theta^t}U_{h+1}^t(s_{h+1})\right]$$

$$+ \sum_{t=1}^{T}\sum_{h=1}^{H}C_0 \cdot \mathbb{E}^{\pi^t}\left[|\mathcal{T}_h^{\pi^t}(r_h^{\theta^t} - r_h^*)(s_h^t, b_h^t)|\right]$$

$$+ \sum_{t=1}^{T}\sum_{h=1}^{H}C_1 \cdot \mathbb{E}^{\pi^t}\left[\mathcal{T}_h^{\pi^t}(r_h^{\theta^t} - r_h^*)^2(s_h^t, b_h^t)\right] \quad \text{(B.5)}$$

Where (a) is from Lemma 9, (b) is by Lemma 10, and Notice that $X_t^h = |\mathcal{T}_h^{\pi^t}(r_h^{\theta^t} - r_h^*)(s_h^t, b_h^t)| \leq 1$, by Lemma 3 (setting $\eta = \frac{1}{2}$), we have

$$\sum_{t=1}^{T}\mathbb{E}^{\pi^t}\left[X_t^h\right] \leq \sum_{t=1}^{T}X_t + \frac{1}{2}\text{Var}^{\pi^t}\left[X_t^h|\mathcal{F}_{t-1}\right] + 2\log\frac{1}{\delta}$$

$$\overset{(a)}{\leq} \sum_{t=1}^{T}X_t^h + \frac{1}{2}\text{Var}^{\pi^t}\left[X_t^h\right] + 2\log\frac{1}{\delta}$$

$$\overset{(b)}{\leq} \sum_{t=1}^{T}X_t^h + \frac{1}{2}\mathbb{E}^{\pi^t}\left[(X_t^h)^2\right] + 2\log\frac{1}{\delta}$$

$$\overset{(c)}{\leq} \sum_{t=1}^{T}X_t^h + \frac{1}{2}\mathbb{E}^{\pi^t}\left[X_t^h\right] + 2\log\frac{1}{\delta},$$

where (a) is by the property of conditional variance; (b) is by $\text{Var}[X] = \mathbb{E}[X^2] - \mathbb{E}[X]^2$; (c) is by the fact that $0 \leq X_t \leq 1$. Hence, we get

$$\sum_{t=1}^{T}\mathbb{E}^{\pi^t}\left[X_t^h\right] \leq 2\sum_{t=1}^{T}X_t^h + 4\log\frac{1}{\delta}.$$

By taking a union bound over all $h \in [H]$, we know for any $h \in [H]$, with probability $1 - \delta$,

$$\sum_{t=1}^{T}\mathbb{E}^{\pi^t}\left[X_t^h\right] \leq 2\sum_{t=1}^{T}X_t^h + 4\log\frac{H}{\delta}.$$

Summing over $h \in [H]$ and considering $X_t^h = |\mathcal{T}_h^{\pi^t}(r_h^{\theta^t} - r_h^*)(s_h^t, b_h^t)|$, we get

$$\sum_{t=1}^{T} \sum_{h=1}^{H} \mathbb{E}^{\pi^t} \left[ |\mathcal{T}_h^{\pi^t}(r_h^{\theta^t} - r_h^*)(s_h^t, b_h^t)| \right] \leq 2 \sum_{t=1}^{T} \sum_{h=1}^{H} |\mathcal{T}_h^{\pi^t}(r_h^{\theta^t} - r_h^*)(s_h^t, b_h^t)| + 4H \log \frac{H}{\delta}.$$

Similarly, we could also get

$$\sum_{t=1}^{T} \sum_{h=1}^{H} \mathbb{E}^{\pi^t} \left[ |\mathcal{T}_h^{\pi^t}(r_h^{\theta^t} - r_h^*)(s_h^t, b_h^t)|^2 \right] \leq 2 \sum_{t=1}^{T} \sum_{h=1}^{H} |\mathcal{T}_h^{\pi^t}(r_h^{\theta^t} - r_h^*)(s_h^t, b_h^t)|^2 + 4H \log \frac{H}{\delta}.$$

Inserting the above result beck to equation (B.5), we have

$$I_2 \leq \sum_{t=1}^{T} \sum_{h=1}^{H} \mathbb{E}^{\pi^t} \left[ (U_h^t)(s_h^t, a_h^t, b_h^t) - \mathbb{T}_{h+1}^{*,\theta^t} U_{h+1}^t(s_{h+1}^t) \right]$$

$$+ \sum_{t=1}^{T} \sum_{h=1}^{H} 2C_0 \cdot \left[ |\mathcal{T}_h^{\pi^t}(r_h^{\theta^t} - r_h^*)(s_h^t, b_h^t)| \right]$$

$$+ \sum_{t=1}^{T} \sum_{h=1}^{H} 2C_1 \cdot \left[ \mathcal{T}_h^{\pi^t}(r_h^{\theta^t} - r_h^*)^2(s_h^t, b_h^t) \right] + O(H \log(H/\delta)).$$

Then using the fact that $|\mathcal{T}_h^{\pi^t}(r_h^{\theta^t} - r_h^*)^2(s_h^t, a_h^t, b_h^t)| \leq |\mathcal{T}_h^{\pi^t}(r_h^{\theta^t} - r_h^*)(s_h^t, a_h^t, b_h^t)|$, we can further have

$$I_2 \leq \sum_{t=1}^{T} \sum_{h=1}^{H} \mathbb{E}^{\pi^t} \left[ (U_h^t)(s_h, a_h, b_h) - \mathbb{T}_{h+1}^{*,\theta^t} U_{h+1}^t(s_{h+1}) \right]$$

$$+ \sum_{t=1}^{T} \sum_{h=1}^{H} 2(C_0 + C_1) \cdot \left[ |\mathcal{T}_h^{\pi^t}(r_h^{\theta^t} - r_h^*)(s_h^t, b_h^t)| \right] + O(H \log(H/\delta)).$$

Furthermore, using decoupling-coefficient assumption 1 with the definition of $d_1$ and $d_2$, we can get

$$I_2 \leq \mu_1 \cdot \sum_{t=1}^{T} \sum_{h=1}^{H} \sum_{i=1}^{t-1} \mathbb{E}^{\pi^i} [(U_h - \mathbb{T}_{h+1}^{*,\theta^i} U_{h+1})(s_h, a_h, b_h)^2] + \frac{d_1}{\mu_1}$$

$$+ 2(C_0 + C_1) \cdot \mu_2 \sum_{t=1}^{T} \sum_{h=1}^{H} \sum_{i=1}^{t-1} [\mathcal{T}_h^{\pi^t}((r_h^{\theta^t} - r_h^*)(s_h^i, b_h^i))^2] + 2(C_0 + C_1) \cdot \frac{d_2}{\mu_2}$$

$$+ O(H \log(H/\delta)).$$

At last, we exploit the Lemma 3 again, and with probability at least $1 - \delta$, we have

$$I_2 \leq \mu_1 \cdot \sum_{t=1}^{T} \sum_{h=1}^{H} \sum_{i=1}^{t-1} \mathbb{E}^{\pi^i} [(U_h - \mathbb{T}_{h+1}^{*,\theta^t} U_{h+1})(s_h, a_h, b_h)^2] + \frac{d_2}{\mu_1}$$

$$+ 4(C_0 + C_1) \cdot \mu_2 \sum_{t=1}^{T} \sum_{h=1}^{H} \sum_{i=1}^{t-1} \mathbb{E}^{\pi^i} [\mathcal{T}_h^{\pi^i}((r_h^{\theta^t} - r_h^*))^2] + 2(C_0 + C_1) \cdot \frac{d_1}{\mu_2}$$

$$+ O(H \log(H/\delta)). \tag{B.6}$$

Now note that $\eta_1 = \eta_2 = 1/\sqrt{T}$, and by choosing $\mu_1 = \frac{\eta_1}{4C_\eta^2}$, $\mu_2 = \frac{\eta_2}{8(C_0+C_1)}$, combining (B.3), and (B.6), with probability at least $1 - 3\delta$, we can have

$$\text{Reg}(T) = I_1 + I_2$$

$$\leq \frac{1}{\sqrt{T}} \cdot HT \cdot (\beta_1 + \beta_2) + \frac{d_1}{\mu_1} + 2(C_0 + C_1) \cdot \frac{d_2}{\mu_2} + O(H \log(H/\delta))$$

$$= \sqrt{T} H(\beta_1 + \beta_2) + 4C_\eta^2 d_1 \sqrt{T} + 16(C_0 + C_1)^2 d_2 \sqrt{T} + O(H \log(H/\delta))$$

$$= \left( H(\beta_1 + \beta_2) + 4C_\eta^2 d_1 + 16(C_0 + C_1)^2 d_2 \right) \sqrt{T} + O(H \log(H/\delta))$$

## C  PROOF OF DECOUPLING COEFFICIENT BOUNDS

We mainly generalize the proof of Proposition 1-3 in Xiong et al. (2022) in this section.

*Proof of Proposition 1.* We first note that the completeness assumption is satisfied in linear MSG case whose proof can be found in Huang et al. (2021); Chen et al. (2023). Now we consider two arbitrary vector $\omega_h, \omega_{h+1} \in \mathbb{R}^d$ whose norms are bounded $H\sqrt{d}$. We define a function $\tilde{U} \in \mathcal{U}$ such that $\tilde{U}_h = \phi^\top \omega_h$ and $\tilde{U}_{h+1} = \phi^\top \omega_{h+1}$. Furthermore more we take arbitrary $\theta = \{\theta_h\}_{h \in H} \subset \mathbb{R}^d$ such that $\|\theta_h\| \leq \sqrt{d}$. Then we could find $r = \{r_h\}_{h \in [H]} \subseteq \mathcal{F}_r$ and $r_h = \phi(s, a, b)^\top \theta_h, \forall h \in [H], (s, a, b) \in \mathcal{S} \times \mathcal{A} \times \mathcal{B}$. Then by Assumption 3, we can find some $U \in \mathcal{U}$ and the corresponding vector $\omega_h(U) \in \mathbb{R}^d$ such that $\|\omega_h(U)\| \leq H\sqrt{d}$ and $\mathbb{T}_h^{*,\theta}(\phi(s, a, b)^\top \omega_{h+1}) = \phi(s, a, b)^\top \omega_h(U) = U_h \in \mathcal{U}_h$. Therefore, we have

$$l_h(\tilde{U}, \theta, s, a, b) = \tilde{U}_h(s, a, b) - \mathbb{T}_h^{*,\theta}(\tilde{U}_{h+1}) = \phi(s, a, b)^\top(\omega_h - \omega_h(U)) = \phi(s, a, b)^\top \Delta_h(U, \tilde{U})$$

where $\Delta_h(U, \tilde{U}) \in \mathbb{R}^d$ and $\|\Delta_h\| \leq 2H\sqrt{d}$.

For any $\{\rho^s\}_{s \in [t]} \subset \varrho_1$, i.e. we take any sequence of the leader and follower's joint policies $\{(\pi^s, \upsilon^{\pi^s, \theta^s})\}_{s \in [t]} \subset \Pi$, we denote as $\phi_h^s = \mathbb{E}^{\rho^s}[\phi(s_h, a_h, b_h)]$ and denote $\Phi_t^h = \lambda I + \sum_{s=1}^t E^{\rho^s}[\phi(s_h, a_h, b_h)\phi(s_h, a_h, b_h)^\top]$, where $\lambda \geq 1$ is a tuning parameter. We further have

$$\mathbb{E}^{\rho^s}[l_h^t(\tilde{U}^t, \theta^t, s_h^t, a_h^t, b_h^t)] - \mu \sum_{s=1}^{t-1} \mathbb{E}^{\rho^s}[l_h^t(\tilde{U}^t, \theta^t, s_h^t, a_h^t, b_h^t)^2]$$

$$= \Delta_h(\tilde{U}^t, U_t)^\top \phi_h^t - \mu \Delta_h(\tilde{U}^t, U_t)^\top \sum_{s=1}^{t-1} \mathbb{E}^{\rho^s}[\phi(s_h^s, a_h^s, b_h^s)\phi(s_h^s, a_h^s, b_h^s)^\top] \Delta_h(\tilde{U}_t, U_t)$$

$$\leq \Delta_h(\tilde{U}_t, U_t)^\top \phi_h^t - \mu \Delta_h(\tilde{U}_t, U_t)^\top \Phi_{t-1}^h \Delta_h(\tilde{U}_t, U_t) + 4\mu\lambda H^2 d$$

$$\leq \frac{1}{4\mu}(\phi_h^t)^\top(\Phi_{t-1}^h)^{-1}\phi_h^t + 4\mu\lambda H^2 d$$

where the first inequality uses Jensen's inequality and $\|\Delta_h(\tilde{U}_t, U_t)\| \leq 2H\sqrt{d}$ and the second inequality exploits the fact that

$$a^\top b \leq (\|a\|_{\Phi_{t-1}^h}\|b\|_{(\Phi_{t-1}^h)^{-1}}) \leq \frac{1}{2}(\|a\|_{\Phi_{t-1}^h}^2 + \|b\|_{(\Phi_{t-1}^h)^{-1}}^2)$$

Summing over $t \in [T]$ and $h \in [H]$, we have

$$\sum_{t=1}^T \sum_{h=1}^H \left(\mathbb{E}^{\rho^s}[l_h(\tilde{U}^t, \theta^t, s_h^t, a_h^t, b_h^t)] - \mu \sum_{s=1}^{t-1} \mathbb{E}^{\rho^s}[l_h(\tilde{U}^t, \theta^t, s_h^t, a_h^t, b_h^t)^2]\right)$$

$$\leq \sum_{h=1}^H \left(\frac{\ln(\det(\Phi_T^h)) - d \ln \lambda}{2\mu} + 4\mu\lambda dH^2 T\right)$$

$$\leq \left(\frac{dH \ln(1 + \frac{T}{d\lambda})}{2\mu} + 4\mu\lambda dH^3 T\right)$$

where the first inequality exploit Lemma 4 and the second inequality uses

$$\ln \det(\Phi_T^h) \leq d \ln \frac{\text{tr}(\Phi_T^t)}{d}, \qquad \text{where } \text{tr}(\Phi_T^h) \leq \lambda d + T$$

By setting $\lambda = \min\{1, \frac{1}{\mu^2 H^2 T}\}$, we have

$$d_1 \leq 2dH(2 + \ln(2HT))$$

Similarly, for $d_2$, notice we could still write

$$m_h(\tilde{\theta}, s, a, b) = r_h^{\tilde{\theta}}(s, b) - r_h(s, b) = \phi(s, a, b)^\top(\tilde{\theta}_h - \theta_h) = \phi(s, a, b)^\top \delta_h(\tilde{\theta}, \tilde{\theta})$$

Then we could repeat the above process to generate the similar bound. Another way to get an upper bound for $d_2$ is to write $r_h^{\bar{\theta}}(s,b) - r_h(s,b)$ as a bilinear form and then use the classical decoupling coefficient results on this class. The readers could see Dann et al. (2021); Chen et al. (2023) for reference.

*Proof of Proposition 2.* We first note that the completeness assumption is also satisfied in generalized linear MSG (Huang et al., 2021; Chen et al., 2023). Similarly, we consider two arbitrary vector $\omega_h, \omega_{h+1} \in \mathbb{R}^d$ whose norms are bounded $H\sqrt{d}$. We define a function $\tilde{U} \in \mathcal{U}$ such that $\tilde{U}_h = \phi^\top \omega_h$ and $\tilde{U}_{h+1} = \phi^\top \omega_{h+1}$. Furthermore more we take arbitrary $\theta = \{\theta_h\}_{h \in H} \subset \mathbb{R}^d$ such that $\|\theta_h\| \leq \sqrt{d}$. Then we could find $r \in \mathcal{F}_r$ and $r_h = \sigma(\phi(s,a,b)^\top \theta_h), \forall h \in [H], (s,a,b) \in \mathcal{S} \times \mathcal{A} \times \mathcal{B}$. Then by Assumption 3, we can find some $U \in \mathcal{U}$ and the corresponding vector $\omega_h(U) \in \mathbb{R}^d$ such that $\|\omega_h(U)\| \leq H\sqrt{d}$ and $\mathbb{T}_h^{*,\theta}(\phi(s,a,b)^\top \omega_{h+1}) = \phi(s,a,b)^\top \omega_h(U) = U_h \in \mathcal{U}_h$. Therefore, we have

$$l_h(\tilde{U}, \theta, s, a, b) = \tilde{U}_h(s,a,b) - \mathbb{T}_h^{*,\theta}(\tilde{U}_{h+1}) = \sigma(\phi^\top \omega_h) - \sigma(\phi^\top \omega_h(U))$$

By the Lipschitz condition we have

$$c_1 \left|\phi^\top \Delta_h(U, \tilde{U})\right| \leq \left|l_h(\tilde{U}, \theta, s, a, b)\right| \leq c_2 \left|\phi^\top \Delta_h(U, \tilde{U})\right|$$

where $\Delta_h(U, \tilde{U}) \in \mathbb{R}^d$ and $\|\Delta_h\| \leq 2H\sqrt{d}$.

For any $\{\rho^s\}_{s \in [t]} \subset \varrho_1$, i.e. we take sequence of $\{\pi^s\}_{s \in [t]} \subset \Pi$, we let $\phi_h^s = \mathbb{E}^{\rho^s}[\phi(s_h, a_h, b_h)]$ and let $\Phi_t^h = \lambda I + \sum_{s=1}^t E^{\rho^s}[\phi(s_h, a_h, b_h)\phi(s_h, a_h, b_h)^\top]$, where $\lambda \geq 1$ is a tuning parameter. We further have

$$\mathbb{E}^{\rho^s}[l_h^t(\tilde{U}^t, \theta^t, s_h^t, a_h^t, b_h^t)] - \mu \sum_{s=1}^{t-1} \mathbb{E}^{\rho^s}\left[l_h^t(\tilde{U}^t, \theta^t, s_h^t, a_h^t, b_h^t)^2\right]$$

$$\leq c_2 \left|\Delta_h(\tilde{U}^t, U_t)^\top \phi_h^t\right| - \mu c_1^2 \Delta_h(\tilde{U}^t, U_t)^\top \sum_{s=1}^{t-1} \mathbb{E}^{\rho^s}\left[\phi(s_h^s, a_h^s, b_h^s)\phi(s_h^s, a_h^s, b_h^s)^\top\right]\Delta_h(\tilde{U}_t, U_t)$$

$$\leq c_2 \Delta_h(\tilde{U}^t, U_t)^\top \phi_h^t - \mu c_1^2 \Delta_h(\tilde{U}^t, U_t)^\top \Phi_{t-1}^h \Delta_h(\tilde{U}_t, U_t) + 4\mu c_1^2 \lambda H^2 d$$

$$\leq \frac{c_2^2}{4\mu c_1^2}(\phi_h^t)^\top (\Phi_{t-1}^h)^{-1}\phi_h^t + 4\mu c_1^2 \lambda H^2 d$$

Summing over $t \in [T]$ and $h \in [H]$, we have

$$\sum_{t=1}^T \sum_{h=1}^H \left(\mathbb{E}^{\rho^s}[l_h(\tilde{U}^t, \theta^t, s_h^t, a_h^t, b_h^t)] - \mu \sum_{s=1}^{t-1} \mathbb{E}^{\rho^s}\left[l_h(\tilde{U}^t, \theta^t, s_h^t, a_h^t, b_h^t)^2\right]\right)$$

$$\leq \sum_{h=1}^H c_2^2 \left(\left(\frac{\ln(\det(\Phi_T^h)) - d\ln\lambda}{2\mu c_1^2} + 4\mu\lambda c_1^2 dH^2 T\right)\right)$$

$$\leq dHc_2^2 \left(\frac{\ln(1 + \frac{T}{d\lambda})}{2\mu c_1^2} + 4\mu c_1^2 \lambda H^2 T\right)$$

By setting $\lambda = \min\{1, \frac{1}{\mu^2 c_1^2 H^2 T}\}$, we have

$$d_1 \leq 2\frac{c_2^2}{c_1^2} dH(2 + \ln(2HT))$$

Similarly, for $d_2$, notice we could still write

$$m_h(\tilde{\theta}, s, a, b) = r_h^{\bar{\theta}}(s,b) - r_h(s,b) = \phi(s,a,b)^\top(\tilde{\theta}_h - \theta_h) = \phi(s,a,b)^\top \delta_h(\tilde{\theta}, \tilde{\theta})$$

Then we could repeat the above process to generate the upper bound. Similarly, another way to get an upper bound for $d_2$ is to exploit Lipschitz condition to upper and lower bound $r_h^{\bar{\theta}}(s,b) - r_h(s,b)$ by two bilinear forms and then use the classical decoupling coefficient results on this class. The readers could see Dann et al. (2021); Chen et al. (2023) for reference.

## D    PROOF OF THEOREM 2

*Proof.* At first, we could decompose the regret into three terms:

$$
\begin{aligned}
\mathrm{Reg}(T) &= \sum_{t=1}^{T} J(\pi^*) - J(\pi^t) \\
&= \underbrace{\sum_{t=1}^{T} \left( \mathbb{E}_{x\sim\rho, a\sim\pi^*}[u^*(x,a)] - \mathbb{E}_{x\sim\rho, a\sim\pi^t}[u^{\theta^t}(x,a)] \right)}_{I_1} \\
&\quad + \underbrace{\sum_{t=1}^{T} \left( \mathbb{E}_{x\sim\rho, a\sim\pi^t}[u^{\theta^t}(x,a)] - \mathbb{E}_{x\sim\rho, a\sim\pi^t}[u^*(x,a)] \right)}_{I_2} \\
&\quad - \underbrace{\sum_{t=1}^{T} \beta \cdot \left( \mathbb{D}_{\mathrm{KL}}(\pi^* \parallel \pi_{\mathrm{ref}}) - (\mathbb{D}_{\mathrm{KL}}(\pi^t \parallel \pi_{\mathrm{ref}})) \right)}_{I_3}.
\end{aligned}
$$

First, we compute the upper bound of $I_1$. By the definition of $\pi^t$ and $\theta^t$, we can get

$$
\mathbb{E}_{x\sim\rho, a\sim\pi^*(\cdot|x)}[u^*(x,a)] - \beta \mathbb{D}_{\mathrm{KL}}[\pi^* \parallel \pi_{\mathrm{ref}}] - \eta_1 L^t(\theta^*)
$$
$$
\leq \mathbb{E}_{x\sim\rho, a\sim\pi^t(\cdot|x)}[u^{\theta^t}(x,a)] - \beta \mathbb{D}_{\mathrm{KL}}[\pi^t \parallel \pi_{\mathrm{ref}}] - \eta_1 L^t(\theta^t),
$$

which is equivalent to

$$
\mathbb{E}_{x\sim\rho, a\sim\pi^*}[u^*(x,a)] - \mathbb{E}_{x\sim\rho, a\sim\pi^t}[u^{\theta^t}(x,a)]
$$
$$
\leq \beta \mathbb{D}_{\mathrm{KL}}[\pi^* \parallel \pi_{\mathrm{ref}}] - \beta \mathbb{D}_{\mathrm{KL}}[\pi^t \parallel \pi_{\mathrm{ref}}] + \eta_1 \cdot \left( L^t(\theta^*) - L^t(\theta^t) \right).
$$

Now we introduce the Lemma 2 and Lemma 4 in Cen et al. (2024) to further bound the cross-entropy loss:

**Lemma 11** (Lemma 2 and 4 in Cen et al. (2024) when $0 \leq R(x,y) \leq 1$). *The following inequality holds with probability at least $1 - \delta$ that*

$$
L^t(\theta^*) - L^t(\theta^t) \leq -(3+e^2)^{-2} \eta^2 \sum_{i=1}^{t-1} \mathbb{E}_{x\sim\rho, a\sim\pi^i} \left[ \left| \delta^*(x^t, a^t) - \delta^t(x^t, a^t) \right|^2 \right] + 2\log\left( \frac{|\mathcal{R}|}{\delta} \right),
$$

*where $\delta^*(x,a) = R^*(x,y_1) - R^*(x,y_2)$, $\delta^t(x,a) = R^{\theta^t}(x,y_1) - R^{\theta^t}(x,y_2)$.*

Then, we compute the upper bound of $I_2$.

$$
\begin{aligned}
I_2 &= \sum_{t=1}^{T} \left( \mathbb{E}_{x\sim\rho, a\sim\pi^t}[u^{\theta^t}(x,a)] - \mathbb{E}_{x\sim\rho, a\sim\pi^t}[u^*(x,a)] \right) \\
&= 2 \sum_{t=1}^{T} \left( \mathbb{E}_{x\sim\rho, y\sim\pi^t}[R^{\theta^t}(x,y)] - \mathbb{E}_{x\sim\rho, y\sim\pi^t}[R^*(x,y)] \right) \\
&\quad - 2 \sum_{t=1}^{T} \left( \mathbb{E}_{x\sim\rho, y\sim\pi_{\mathrm{base}}}[R^{\theta^t}(x,y)] - \mathbb{E}_{x\sim\rho, y\sim\pi_{\mathrm{base}}}[R^*(x,y)] \right) \\
&\leq 2 \sum_{t=1}^{T} \left( \mathbb{E}_{x\sim\rho, y_1\sim\pi^t, y_2\sim\pi_{\mathrm{base}}}[\delta^t(x,y_1,y_2) - \delta^*(x,y_1,y_2)] \right).
\end{aligned}
$$

By Multi-agent Decoupling Coefficient, we can further derive

$$I_2/2 \leq \mu \cdot \sum_{t=1}^{T} \sum_{i=1}^{t-1} \left( \mathbb{E}_{x \sim \rho, y_1 \sim \pi^i, y_2 \sim \pi_{\text{base}}} [(\delta^t(x, y_1, y_2) - \delta^*(x, y_1, y_2))^2] \right) + \frac{d}{4\mu}$$

$$\leq \mu \cdot \sup_{x,y,i} \frac{\pi_{\text{base}}(y \mid x)}{\pi^i(y \mid x)} \cdot \sum_{t=1}^{T} \sum_{i=1}^{t-1} \left( \mathbb{E}_{x \sim \rho, y_1 \sim \pi^i, y_2 \sim \pi^i} [(\delta^t(x, y_1, y_2) - \delta^*(x, y_1, y_2))^2] \right) + \frac{d}{4\mu}$$

$$= \mu \cdot \sup_{x,y,i} \frac{\pi_{\text{base}}(y \mid x)}{\pi^i(y \mid x)} \cdot \sum_{t=1}^{T} \sum_{i=1}^{t-1} \left( \mathbb{E}_{x \sim \rho, a \sim \pi^i} [(\delta^t(x, a) - \delta^*(x, a))^2] \right) + \frac{d}{4\mu}.$$

Note that

$$\frac{\pi_{\text{base}}(y \mid x)}{\pi^i(y \mid x)} = \frac{\pi_{\text{base}}(y \mid x)}{\pi_{\text{ref}}(y \mid x)} \cdot \frac{\pi_{\text{ref}}(y \mid x)}{\pi^i(y \mid x)} = \kappa \cdot \frac{\pi_{\text{ref}}(y \mid x)}{\pi^i(y \mid x)}$$

Then by $\pi^i(y \mid x) \propto \pi_{\text{ref}}(y \mid x) \exp(R^i(x, y)/\beta)$ in Rafailov et al. (2024), we can derive $|\log \pi^i(y \mid x) - \log \pi^{\text{ref}}(y \mid x)| \leq 2\|R^i(x, \cdot)/\beta\|_\infty \leq 2/\beta$ (Cen et al. (2022), Appendix A.2), then $\frac{\pi_{\text{ref}}(y|x)}{\pi^i(y|x)} \leq \exp(2/\beta)$. Then

$$\sup_{x,y,i} \frac{\pi_{\text{base}}(y \mid x)}{\pi^i(y \mid x)} = \kappa \exp(2/\beta).$$

Now we sum over $I_1, I_2$ and $I_3$. Thus, we can get

$$\text{Reg}(T) = I_1 + I_2 + I_3$$

$$= \sum_{t=1}^{T} \left( \eta_1 \cdot (L^t(\theta^*) - L^t(\theta^t)) \right) + I_2$$

$$\leq -(3 + e^2)^{-2} \eta_1 \cdot \eta^2 \cdot \sum_{t=1}^{T} \sum_{i=1}^{t-1} \mathbb{E}_{x \sim \rho, a \sim \pi^i} \left[ |\delta^*(x^t, a^t) - \delta^t(x^t, a^t)|^2 \right] + 2\eta_1 T \log \left( \frac{|\mathcal{R}|}{\delta} \right)$$

$$+ 2\mu \cdot \kappa \cdot \exp(2/\beta) \cdot \sum_{t=1}^{T} \sum_{i=1}^{t-1} \left( \mathbb{E}_{x \sim \rho, a \sim \pi^i} [(\delta^t(x, a) - \delta^*(x, a))^2] \right) + \frac{d}{2\mu}.$$

Now we choose $\eta_1 = 2\mu\kappa \exp(2/\beta) \cdot (3 + e^2)^2 \cdot \eta^{-2} = 1/\sqrt{T}$, then the inequality above will become

$$\text{Reg}(T) \leq 2\sqrt{T} \log \frac{|\mathcal{R}|}{\delta} + 2 \cdot (3 + e^2)^2 \eta^{-2} d\kappa \exp(2/\beta) \sqrt{T}.$$

$\square$

