# OpenReview forum: "Exploration in the Face of Strategic Responses: Provable Learning of Online Stackelberg Games"
_ICLR.cc/2025/Conference — Submitted to ICLR 2025_

### Official Review · Reviewer_Hm8r · 2024-10-28

**Soundness:** 3
**Presentation:** 3
**Contribution:** 2
**Rating:** 5
**Confidence:** 3

**Summary:**

This paper studies online leader-follower Markovian games where a leader interacts with a myopic follower using a quantal response policy. The leader lacks knowledge of the follower's reward function, posing a challenge in learning through strategic interactions. The authors introduce the PES algorithm, which estimates the leader’s value function and the follower’s response mapping by maximizing the sum of the Bellman error of the value function, the likelihood of the quantal response model, and an additional regularization term. This approach allows for policy updates and achieves sublinear regret, avoiding complex optimistic planning. Finally, the authors show how their algorithm applies to RLHF problems.

**Strengths:**

- The problem formulation is clear, and I believe it could be of interest to the RL community.
- The authors propose a possible application of their algorithm to RLHF settings, which is currently a hot topic. Thus, this represents a strength.

**Weaknesses:**

- At line 149, the authors refer to a specific communication layer in Markovian Stackelberg games, but I believe it should be called "commitment layer".
- It is a bit confusing in the initial section when the authors say, "we use the notion of sample complexity", and then they provide a regret upper bound.
- Given the current version of the paper, I cannot evaluate the theoretical guarantees of the algorithm proposed in the paper, as the authors do not provide any discussion about the tightness of their regret upper bound.
- Furthermore, several lemmas are taken from previous papers. The authors should better outline the technical steps to derive their results.

**Questions:**

- The use of entropy to model the followers' bounded rationality is something that exists in the literature? Can you provide additional related works that employ the same approach?
- The paper shares several features with (Chen et al., 2023). Could the authors clarify the main differences between this work and theirs?
- Is the regret upper bound tight?

---

> ### Author Response · Authors · 2024-11-30
>
> We thank the reviewer for taking the time to read our paper, and for the comments and suggestions. Here are our main responses:
> ### Q1: The use of entropy to model the followers' bounded rationality is something that exists in the literature? Can you provide additional related works that employ the same approach?
>
> In our knowledge, the definition of bounded rationality is extensively used in the context of Stackelberg Games([1], [2],[3]) and real-world applications ([4], [5].
>
> ### Q2: The paper shares several features with (Chen et al., 2023). Could the authors clarify the main differences between this work and theirs?
>
> The main difference between our work and Chen's work in [6] is the planning step:
> Chen's work first construct a confidence set and then conduct the optimistic planning step, which induced a *complicated constraint optimization problem*, while we conduct the planning step by minimizing our regularized loss function, which induced an *unconstraint optimization problem*.
>
> The features we shared with Chen's work in [6], are mainly about their *performance difference lemma*, which, as they stated in their introduction section, relates the suboptimality of the learned policy to the Bellman error of the upper level problem and the estimation error of the response model in the lower level problem and *might be of independent interest*. In our paper, we exploit this lemma to decompose the suboptimality of the learned policy into the corresponding two terms errors. However, the way we bound those two terms of errors and achieve a sub-linear regret is different.
>
> ### Q3: Is the regret upper bound tight?
> Our upper bound does match the lower bound on $T$ with an extra logarithmic factor $\log T$. For the proof of showing $O(\sqrt{T})$ is the lower bound for online learning problem, please refer to [7].
>
> Our bound may be not tight in the decoupling coefficients $d$. Making improvement on these terms is highly non-trivial. A potential way to achieve this is to exploit advanced techniques such as variance reduction. We would like to leave this improvement as our future work.
>
> 1. Černý, Jakub, et al. "Computing quantal stackelberg equilibrium in extensive-form games." Proceedings of the AAAI Conference on Artificial Intelligence. Vol. 35. No. 6. 2021.
> 2. Rong Yang, Fernando Ordonez, and Milind Tambe. Computing optimal strategy against quantal response in security games. In AAMAS, pages 847–854, 2012.
> 3. Arunesh Sinha, Debarun Kar, and Milind Tambe. Learning adversary behavior in security games: A pac model perspective. arXiv preprint arXiv:1511.00043, 2015.
> 4. Fei Fang, Thanh H Nguyen, Rob Pickles, Wai Y Lam, Gopalasamy R Clements, Bo An, Amandeep Singh, Brian C Schwedock, Milin Tambe, and Andrew Lemieux. Paws—a deployed game-theoretic application to combat poaching. AI Magazine, 38(1):23–36, 2017.
> 5. Bo An, Fernando Ordóñez, Milind Tambe, Eric Shieh, Rong Yang, Craig Baldwin, Joseph DiRenzo III, Kathryn Moretti, Ben Maule, and Garrett Meyer. A deployed quantal response- based patrol planning system for the us coast guard. Interfaces, 43(5):400–420, 2013.
> 6.  Siyu Chen, Mengdi Wang, and Zhuoran Yang. Actions speak what you want: Provably sampleefficient reinforcement learning of the quantal stackelberg equilibrium from strategic feedbacks.arXiv preprint arXiv:2307.14085, 2023.
> 7.  Auer, Peter, Thomas Jaksch, and Ronald Ortner. "Near-optimal regret bounds for reinforcement learning." Advances in neural information processing systems 21 (2008).

---

### Official Review · Reviewer_ryQi · 2024-10-29

**Soundness:** 3
**Presentation:** 3
**Contribution:** 3
**Rating:** 6
**Confidence:** 3

**Summary:**

The paper studies the online stackelberg game, where a leader and a myopic follower interact with the environment and receive feedback. The paper proposed a Planning After Estimation algorithm and showed its regret bound. The authors also applied the algorithm to RLHF framework, exhibiting its potential application.

**Strengths:**

Compared to Chen et al. (2023), the PES algorithm is easy to implement with rigorous theoretical guarantees. The PES algorithm is fairly easy to understand with two steps. The algorithm is also applicable to the RLHF framework, which shows a potential nice application to real-world problems.

**Weaknesses:**

However, I feel that the assumption of bounded rationalness is quite restricted. From my understanding, the follower takes the best strategy with an extra regularized term, assuming that the follower has the knowledge of the environment such as the reward function. Does this assumption make sense in real-world application? What if the follower is generally unaware of the environment and has to learn from scratch?

**Questions:**

Can the online stackelberg game be formulated as a pure single-agent RL game? It seems that the quantal response model of the follower can be treated as one part of the environment. If so, why cannot traditional single-agent RL algorithms be applied to the game?

---

> ### Author Response · Authors · 2024-11-30
> **Response to the Weakness Part**
>
> We thank the reviewer for taking the time to read our paper, and for the comments and suggestions. Here are our main responses:
> ### W1: Does the bounded rational assumption make sense in real-world application?
>
> In our knowledge, the definition of bounded rationality is extensively used in the context of Stackelberg Games([1], [2]), [3] and real-world applications ([4],[5]).
>
> ### W2: What if the follower is generally unaware of the environment and has to learn from scratch?
>
> It's a highly non-trivial question because the main difficulty in learning Stackelber games comes from the follower's reponse model. If the response model changes during learning phase (since the follower will update his/her estimation on his/her reward), the problem becomes completely different and much more challenging. Here we provide some literature on this branch for comparison.
>
> In general, learning in repeated general-sum Stackelberg games remains an open problem [6]. Recently, there are some works focusing on sub-classes of games with specific reward structures. [7] and [8] study zero-sum stochastic Stackelberg games where the sum of rewards of the two players is zero. [9] and [10] study cooperative Stackelberg games where the leader and the follower share the same reward. Also [11] proposed a sample efficient learning algorithm for general-sum Stackelberg games in bandit setting with the assumption that both the follower and the leader only have access to noisy bandit feedback of the realized reward.
>
> On the other hand, our work mainly focuses on learning from the leader’s perspective, assuming access to an oracle of the follower’s best response. Such setting is extensively used in the literature and have real-world applications, as we stated in section 2 of our paper.
> Learning Stackelberg equilibrium in the case when the follower is unware of the environment is an interesting and challenging question but seems to be diversed from our current work. We would like to leave this extension as our future work.
>
> 1. Černý, Jakub, et al. "Computing quantal stackelberg equilibrium in extensive-form games." Proceedings of the AAAI Conference on Artificial Intelligence. Vol. 35. No. 6. 2021.
> 2. Rong Yang, Fernando Ordonez, and Milind Tambe. Computing optimal strategy against quantal response in security games. In AAMAS, pages 847–854, 2012.</a></p>
> 3. Arunesh Sinha, Debarun Kar, and Milind Tambe. Learning adversary behavior in security games: A pac model perspective. arXiv preprint arXiv:1511.00043, 2015.
> 4. Fei Fang, Thanh H Nguyen, Rob Pickles, Wai Y Lam, Gopalasamy R Clements, Bo An, Amandeep Singh, Brian C Schwedock, Milin Tambe, and Andrew Lemieux. Paws—a deployed game-theoretic application to combat poaching. AI Magazine, 38(1):23–36, 2017.
> 5. Bo An, Fernando Ordóñez, Milind Tambe, Eric Shieh, Rong Yang, Craig Baldwin, Joseph DiRenzo III, Kathryn Moretti, Ben Maule, and Garrett Meyer. A deployed quantal response- based patrol planning system for the us coast guard. Interfaces, 43(5):400–420, 2013.
> 6. Yu, Yaolong, and Haipeng Chen. "Decentralized Online Learning in General-Sum Stackelberg Games." arXiv preprint arXiv:2405.03158 (2024).
> 7. Goktas, Denizalp, Sadie Zhao, and Amy Greenwald. "Zero-sum stochastic stackelberg games." Advances in Neural Information Processing Systems 35 (2022): 11658-11672.
> 8. Sun, Jingrui, Hanxiao Wang, and Jiaqiang Wen. "Zero-sum Stackelberg stochastic linear-quadratic differential games." SIAM Journal on Control and Optimization 61.1 (2023): 252-284.
> 9. Kao, Hsu, Chen-Yu Wei, and Vijay Subramanian. "Decentralized cooperative reinforcement learning with hierarchical information structure." International Conference on Algorithmic Learning Theory. PMLR, 2022.
> 10. Geng Zhao, Banghua Zhu, Jiantao Jiao, and Michael Jordan. Online learning in stackelberg games with an omniscient follower. In International Conference on Machine Learning, pp. 42304–42316. PMLR, 2023.
> 11. Yu Bai, Chi Jin, Huan Wang, and Caiming Xiong. Sample-efficient learning of stackelberg equilibria in general-sum games. Advances in Neural Information Processing Systems, 34:25799–25811,2021.

---

> ### Author Response · Authors · 2024-11-30
> **Response to the Question Part**
>
> ### Q1: Can the online stackelberg game be formulated as a pure single-agent RL game? If so why can't we use traditional single-agent RL algorithms to solve this game?
>
> There are two main obstacles for us to to formulate online stackelberg game into a single-agent RL game:
> * The followers (or the quantal response model) have the knowledge of the leader's announced policy $\pi_h$ instead of the bandit action $a_h$. If we treat the follower as the part of the environment, then the transitions will depend on the leader's policy, i.e. assume the kernel in the form of $P(s^\prime|s, \pi, a)$, which is not a common way to model the transitions and the complexity of such model is much higher than the case when the environment is only aware of the action.
> * Even if we circumvent the previous challenge by assuming the follower would only be aware of the action, the kernel or the leader's reward function will still have nested structures with respect to $(s, a)$. Specifically, the reward and the kernel will have the form of $R(s, a) = r(s, a, \mathrm{quantal\\_response}(s, a)$) and $P(s^\prime |s, a) = p(s^\prime| s, a,\mathrm{quantal\\_response}(s, a)$. Such a strcuture may cause more challenges on the estimation even when $r$ and $p$ is a simple function, which is due to the complexity of quantal response function.
> There also exists some work ([12]) that formulates repeated stackelberg game into a single-agent RL game by neglecting the leader-follower structure. In [12], they didn't estimate the response model and just reduce the whole problem into a lipschitz bandit problem. However, their regreat bound (see theorem 1 and 2 in [12]) has much worse dependence on $T$ than ours (i.e. $\tilde{\mathcal{O}}(\sqrt{T})$).
>
> 12. Zhu, Banghua, et al. "The sample complexity of online contract design." arXiv preprint arXiv:2211.05732 (2022).2021.

---

### Official Review · Reviewer_2PX9 · 2024-11-03

**Soundness:** 3
**Presentation:** 2
**Contribution:** 2
**Rating:** 6
**Confidence:** 4

**Summary:**

This paper investigates an online learning problem focused on the leader’s optimal policy in an episodic Markov Stackelberg game. In each round, the leader announces a policy, prompting the follower to respond based on a myopic quantal response strategy. The leader then observes the resulting trajectory and receives bandit feedback, even though both the reward functions and transition model are unknown beforehand. To address this, the authors propose an algorithm, Planning after Estimation, which learns the follower’s reward function and the leader’s value function through function approximation. The algorithm operates by first estimating the leader’s value function and the follower’s response model, followed by a planning step to refine the leader's policy. This approach is efficient, simpler to implement, and achieves sublinear regret under certain assumptions. Furthermore, the paper analyzes the algorithm’s application in a special case relevant to Reinforcement Learning with Human Feedback (RLHF).

**Strengths:**

1. Originality:  The PES algorithm is a fresh approach to solve Stackelberg equilibria in online settings with general function approximation, emphasizing ease of implementation.
2. Quality: Extensive proofs are presented for sublinear regret and computational feasibility, underlining the method's theoretical robustness.
3. Clarity: This paper presents a clean and implementable algorithm.
4. Significance: It  solves the proposed online learning problem.
5. Application to RLHF: Extending PES to an RLHF context demonstrates its practical relevance and adaptability in settings beyond purely theoretical constructs.

**Weaknesses:**

1. Contribution: Currently I do not assess the contribution of this work as very strong. As there are existing works on the same problem, to better clarify the significance of this paper, it is necessary to present a quantitative comparison between previous results and the proposed algorithm, but it seems missing in the paper.
Although the regret has a $\tilde{O}(\sqrt{T})$ dependence on $T$, this result is somewhat restrictive: (1) The regret depends on the decoupling coefficients and covering numbers, which may further depend on $H$ and $T$, while the typical value of covering numbers is not discussed; (2) The algorithm implicitly assumes that $\ita$ is known to the leader; (3) It is based on a function approximation approach, requiring corresponding assumptions for tractability.
Moreover, the result seems less meaningful for the turn-based special case, which includes the presented case study on RLHF model. This is because when the follow er's response only depends on the leader's action instead of mixed strategy, the leader is essentially facing a standard Markov decision process, which is already well-studied.
2. Writing: The writing is acceptable but could be improved in some parts.
(1) The abstract and most early parts of the introduction focus on explaining Stackelberg games, but do not clarify that the model involves a Markov state transition, which is part of the main challenge. (2) The definition of sample complexity in the preliminary section is redundant, as it is not used.
(3) Some formulas need more explanation, such as (3.1) $\nu_h$, (4.8) the meaning of the substracted term, (6.1) $\beta$.
(4) Line34 + " interactions can be sequential..."  Unclear, especial with the use of "can" Does it may the authors were not sure of their use here
(5)cLine 42+ The use of "her" is not clear. Whose gender is female here?

**Questions:**

1. As mentioned in the related work section, some existing works have studied the considered problem and the similar problem where the follower plays the best response instead of a quantal response. Please discuss how the effectiveness of your algorithm compares with these existing results, especially regarding the regret bounds.
2. What is the technical motivation for the first term $W_1^{U,\theta}(s_1)$ in the loss function?
3. The results focus on a general function approximation approach, where restrictive assumptions are required. If the Markov Stackelberg Game has finite state and action spaces, could you derive a regret bound for your algorithm that only depends on $H,T,|S|,|A|,|B|$ and $||u||,||r||$?

---

> ### Author Response · Authors · 2024-11-30
> **Response to the Weakness part.**
>
> We thank the reviewer for taking the time to read our paper, and for the comments and suggestions. Here are our main responses to the weakness part:
> ### Weakness on Writing
> Thank you for your constructive feedback on the writing and clarity of our paper. We accept all the advice and appreciate your attention to detail and your suggestions for improvement.
> For clarity, we intended to use 'his' to refer to the follower and 'her' to refer to the leader. Such reference appears in some literatures, but we did not make any statement about it and we are sorry about that.
> ### W1: The regret depends on the decoupling coefficients and covering numbers, which may further depend on $H$ and $T$, while the typical value of covering numbers is not discussed.
> * **Regarding decoupling coefficients:** We discussed their values in the context of linear and generalized linear cases in Section 5.2, but we will provide more examples and special cases in the revised paper. We agree that we could have provided a more comprehensive discussion about covering numbers. Thank you for highlighting this gap. In the revised version, we will include a detailed explanation of covering numbers and provide typical values for certain scenarios.
> * **Regarding potential dependence on $H$ and $T$**:
>     * **Covering numbers:** Since the parameter set we constructed in Lemma 7 and 8 is not relevant to $H$, so the covering numbers do not have neither explicit nor implicit dependence on $H$. We only need to pay an extra $\log H$ factor due to union bound, which is aleady in the formulae of $\beta_1, \beta_2$. As for $T$, it only shows up in the error tolerance of distance, so the dependence on $T$ is poly-logarithmic at most.
>     * **Decoupling coefficients:** The dependence of these coefficients on $H$ and $T$ is more complex in the general case. However, the specific cases we discussed in Section 5.2 exhibit a linear dependence on $H$ and a logarithmic dependence on $T$.
>
> We appreciate your valuable feedback on this point and will ensure these clarifications will be thoroughly addressed in the revised version.
>
> ### W2: The algorithm implicitly assumes that $\eta$ is known.
> Yes, we indeed assume $\eta$ from the response model is known by the leader, because different $\eta$ represent different followers' behaviors, which leads to different learning problem. Please see our general response for a more detailed disccusion on the setting.
>
> ### W3: It is based on a function approximation approach, requiring corresponding assumptions for tractability.
> We admit this function approximation approach cannot be circumvented in our work. However, if we want to exploit statistical tools in our proof, assuming the true function is from some space seems to be inevitable.
> We could address this term by adding a term of misspecification error and remove the assumption that the true model is within our parameter space.
> ### W4: When the follower's response only depends on the leader's action instead of mixed strategy, the leader is essentially facing a standard Markov decision process.
> We respectfully disagree with this assertion.
>
> If we assume follower's response only depends on the leader's action, the kernel and the leader's reward function will have nested structures with respect to $(s, a)$. Specifically, the reward and the kernel will have the form of $R(s, a) = r(s, a, \mathrm{quantal\\_response}(s, a)$) and $P(s^\prime |s, a) = p(s^\prime| s, a,\mathrm{quantal\\_response}(s, a)$. Such a strcuture may cause more challenges on the estimation even when $r$ and $p$ is a simple function, which is due to the complexity of quantal response function.
>
> There also exists some work [1] that formulates repeated stackelberg game into a single-agent RL game by neglecting the leader-follower structure. In [1], the authors didn't estimate the response model and just reduce the whole problem into a lipschitz bandit problem. However, their regreat bound (see theorem 1 and 2 in [1]) has much worse dependence on $T$ than ours (i.e. $\tilde{\mathcal{O}}(\sqrt{T})$).
>
> Therefore, reducing this case into standard MDP is a highly challenging and non-trival work and can't be done easily. Moreover, current work on this side does not have comparable results than ours.
>
> 1. Zhu, Banghua, et al. "The sample complexity of online contract design." arXiv preprint arXiv:2211.05732 (2022).

---

> ### Author Response · Authors · 2024-11-30
> **Response to the Question 1 and 2**
>
> ### Q1: As mentioned in the related work section, some existing works have studied the considered problem and the similar problem where the follower plays the best response instead of a quantal response. Please discuss how the effectiveness of your algorithm compares with these existing results, especially regarding the regret bounds.
> We would like to clarify that comparing our algorithm, which incorporates a quantal response model, to models where the follower plays a best response is not straightforward, as they address fundamentally different problem structures determined by the agent response model.
>
> In particular, the best response model introduces significant challenges. As shown in [2] (Theorem 1), recovering the reward function of the follower in a best response model is fundamentally ill-posed, which means a slight estimation error in $r$ would result in a complete different best response action. Therefore, in this case, even if the leader could have a good estimation on the follower's reward function, the corresponding estimated response model may still be very different with the true model and such gap would corrupt our policy.
> This issue has led some existing works to assume additional structural conditions that make it possible to learn the reward function.
> For instance:
> * [3] have worked on this issue by introducing binary search techniques.
> * [4] explored a similar approach using binary search.
>
> In these cases, the regret bounds typically grow as $O(\sqrt{T})$, which contrasts with the results we obtain with the quantal response model, where the structure of the problem allows for different types of regret analysis.
> Alternatively, there is a different line of work that does not estimate the response model at all, and instead treats the problem as a single-agent problem. For instance, [1] takes this approach, effectively neglecting the follower’s response structure and reducing the problem to a Lipschitz bandit problem.
>
> As a result, the regret bounds in this case are much worse than $O(\sqrt{T})$.
> We will update the manuscript to more clearly explain these differences and contextualize our approach within these existing results. The revised version will provide a more in-depth discussion of the differences between best response models and quantal response models, especially regarding how they affect the regret bounds and overall problem structure.
>
> ### Q2: What is the technical motivation for the first term $(W_1^{U, \theta}(s_1))$ in the loss function?
> Here, $W_1^{U, \theta}(s_1)$ represents the value of the leader's optimal policy for the estimated model $(U, \theta)$.
> Intuitively, $W_1^{U, \theta}(s_1)$ is a surrogate or estimation of the true value $J(\pi^\star)$ and maximizing $W_1^{U, \theta}(s_1)$ will help us minimizing the sub-optimality of the leader's value. To better illustrate this point, let's first recall the formula of regret:
>
> \begin{align*}
>     \mathrm{Reg}(T) &= \sum_{t=1}^T(J(\pi^\star) - J(\pi^t))\\\\
>         &= \sum_{t=1}^T(W_1^{U^\star, \theta^\star}(s_1) - W_1^{\pi_t}(s_1)).
> \end{align*}
>
> Here, since we don't know the value of $(U^\star, \theta^\star)$ and only have access to $(U^t, \theta^t)$, we insert extra terms $W_1^{U_t, \theta_t}(s_1)$ and have
>
> \begin{align*}
>     \mathrm{Reg}(T) &= \underbrace{\sum _ {t=1}^T(W _ 1^{U^\star, \theta^\star}(s _ 1) - W _ 1^{U^t, \theta^t}(s _ 1))} _ {I _ 1} - \underbrace{\sum _ {t=1}^T(W _ 1^{U^t, \theta^t}(s _ 1) - W _ 1^{\pi _ t}(s _ 1))} _ {I _ 2}.
> \end{align*}
>
> Here, $I_1$ performs like the model approximation error, which could be upper bounded by the difference of two Loss functions.
> In the mean while, $I_2$ is the sub-optimality of the leader's value for the model $(U^t, \theta^t)$, we exploit the performance difference lemma to decompose this term into the leader's Bellman error and  the estimation error of the follower's response model. At last, by selecting right value of the regularizing coefficient (i.e. $\eta$), these two terms of error would be cancelled out by $I_1$.
>
> 1. Zhu, Banghua, et al. "The sample complexity of online contract design." arXiv preprint arXiv:2211.05732 (2022).
> 2. Yu Bai, Chi Jin, Huan Wang, and Caiming Xiong. Sample-efficient learning of stackelberg equilibria in general-sum games. Advances in Neural Information Processing Systems, 34:25799–25811,2021.
> 3. Wu, Jibang, et al. "Contractual reinforcement learning: Pulling arms with invisible hands." arXiv preprint arXiv:2407.01458 (2024).
> 4. Scheid, Antoine, et al. "Incentivized Learning in Principal-Agent Bandit Games." arXiv preprint arXiv:2403.03811 (2024).

---

> ### Author Response · Authors · 2024-11-30
> **Response to  Question 3**
>
> ### Q3: If the Markov Stackelberg Game has finite state and action spaces, could you derive a regret bound for your algorithm that only depends on $H, T, |S|, |A|, |B|$ and $\lVert u\rVert, \lVert r \rVert$?
> Since our proof process is highly relevant to the norm bound of $u$ and $r$, we assume $\lVert u\rVert_{\sup}, \lVert r\rVert_{\sup} \leq 1$, which is common in this field and without loss of generality.
> We first derive explicit formulae for the constants in eqn 5.2 of our paper:
>
> \begin{align*}
> B_A &= 2(\frac{\log |B|}{\eta} + 1),\\\\
> C_\eta &= \mathcal{O}(\frac{1}{\eta} + B_A) = \frac{2}{\eta}(\log |B| +1 + \frac{1}{\eta}),\\\\
> C_0 &= 2\eta H,\\\\
> C_1 &= 2\eta^2H\exp(4\eta)|B|^4.
> \end{align*}
>
> Then we bound the covering numbers in Lemma 7 and Lemma 8:
>
> \begin{align*}
> \mathcal{N} _ {\rho_1}(\Theta, 1/T) &\leq ((1 + \eta)T)^{|S|\cdot |A| \cdot |B|},\\\\
> \mathcal{N} _ {\rho_2}(\mathcal{F}, 1/T) &\leq T^{|S|^2\cdot |A|^2 \cdot |B|^2},\\\\
> \beta _ 1 &= 2 \log (H \cdot \frac{\mathcal{N} _ {\rho _ 1}(\Theta, 1/T)}{\delta}) + 8 \\\\
> &= 2 \log\frac{H}{\delta} + 2 |S|\cdot |A| \cdot |B| \log T + 8,\\\\
> \beta _ 2 &= 4 H^2\log (H \cdot \frac{\mathcal{N} _ {\rho _ 2}(\Theta, 1/T)}{\delta}) + 5 \\\\
> &= 4H^2(\log\frac{H}{\delta} + |S|^2\cdot |A|^2 \cdot |B|^2 \log T) + 5.
> \end{align*}
>
> Then we bound the decoupling coefficient. Notice that the tabular case could be covered by the linear case with dimension $d = |S| \cdot |A| \cdot |B|$, so by the proposition 1 in our paper, we have
> \begin{align*}
> d_1, d_2 &\leq 2|S||A||B|H \cdot (2 + \log (2HT)).
> \end{align*}
>
> At last we combine eveything together to get the explicit formula:
> \begin{align*}
>     Reg(T) &\leq (H(\beta_1 + \beta_2) + 4 C_\eta^2 d_1 + 16(C_0 + C_1)^2 d_2)\sqrt{T} + O(H\log (H/\delta))\\\\
>     &=O(H^3(\log\frac{H}{\delta} + |S|^2\cdot |A|^2 \cdot |B|^2 \log T)\sqrt{T})\\\\
>     &\qquad +O(\frac{1}{\eta^4}\cdot H\log H\cdot |S|\cdot|A|\cdot|B|(\log|B|)^2\cdot \sqrt{T}\log T))\\\\
>     & \qquad + O(\eta^4\exp(8\eta)H^3\log H\cdot |S|\cdot|A|\cdot|B|^9\cdot\sqrt{T}\log T)\\\\
>     &= O((\eta^4 + \frac{1}{\eta^4})\exp(8\eta) \cdot H^3\cdot |S|^2\cdot |A|^2 \cdot |B|^9\cdot \sqrt{T}\cdot\log \frac{HT}{\delta}\cdot (\log |B|)^2).
> \end{align*}
>
> We would like to make a few comment on the derived bound:
> * Our model has a exponential dependence on the regularized coefficient $\eta$, which comes from the performence difference lemma in [5]. When $\eta $ approach to infinity, the quantal reponse model also approach to best response model. Such dependence also reflects the ill-poseness of the best response model.
> * The bound also have a higher-order dependence on $|B|$, which is also from the performence difference lemma in [5]. Such dependence reflects that recovering the follower's reward function (and also the response model) is indeed a challenging work.
> * Our results does match the lower bound on $T$ with an extra logarithmic term $\log(T)$.
>
> We don't rule out the possibility to get better dependence on $\eta$ and $|B|$, but such improvement may need us to find a better technique than performance difference lemma to bound the sub-optimality, which is highly-nontrivial and challenging. We would like to leave such extension as future work.
>
> 5. Siyu Chen, Mengdi Wang, and Zhuoran Yang. Actions speak what you want: Provably sampleefficient reinforcement learning of the quantal stackelberg equilibrium from strategic feedbacks.arXiv preprint arXiv:2307.14085, 2023.

---

### Official Review · Reviewer_jRmd · 2024-11-03

**Soundness:** 3
**Presentation:** 3
**Contribution:** 2
**Rating:** 5
**Confidence:** 3

**Summary:**

The paper studies an online Stackelberg game from the leader's perspective. The leader interacts with a follower who behaves according to a quantal response model. The leader does not know her own utilities, nor does she know the follower's. However, the leader observes the interaction history and attemps to learn the follower's behavior model based on this observation. The aim is to learn an optimal policy to use against the follower. It is assumed that the follower is myopic. And the paper proposes to use a class of parameterized functions to approximate the follower's reward function, whereby the task reduces to estimating the parameter that minimizes the loss with respect to samples collected. Based on the estimation, a greedy planning algorithm is proposed and the authors analyzed the regret of this algorithm.

**Strengths:**

The paper uses some nice ideas, which are novel in the literature on learning in Stackelberg games, such as using function approximation for learning the follower's response model. The authors also made an effort to make the model more realistic by assuming, e.g., the leader doesn't know the follower's realized rewards. The paper also made an interesting connection with RLHF.

**Weaknesses:**

The key contribution is somewhat unclear and the overall results look incremental. Although the authors have put significant good effort into deriving the results, the outcomes look overall like a string of engineering work without strong key insights. It is therefore unclear how much implications the work offers to similar problems and to the literature in general. While the paper made some realistic assumptions, this is weakened by some others, e.g., the follower is myopic. The paper also seems to miss important related works in the literature on Stackelberg games, e.g.:

- Inverse Game Theory for Stackelberg Games: the Blessing of Bounded Rationality, Wu et al., 2022.
- Learning and Approximating the Optimal Strategy to Commit To, Letchford et al., 2009.
- Learning optimal strategies to commit to, Peng et al., 2019.
- Computing optimal strategy against quantal response in security games, Yang et al., 2012.

Typos and minor issues:
- Abstract: First sentence looks confusing at first glance, as if the leader is using a quantal response policy.
- Equation (3.3): I suppose $\pi^*$ should be $\pi^\star$.
- Line 219: "Let ... to be" --> "Let ... be".
- Assumption 1: double parentheses seem unintended.

**Questions:**

How far away are the proposed techniques from addressing the same problem but with a far-sighted follower? What are the main obstacles?

---

> ### Author Response · Authors · 2024-11-30
>
> We thank the reviewer for taking the time to read our paper, and for the comments and suggestions. Here are our main responses:
> ### Q1: Lack of comparison with some classic literatures in Stackelberg games.
> Thank you for pointing out the omission of important related works in the literature on Stackelberg games. We greatly appreciate your feedback, as it allows us to improve the comprehensiveness of our manuscript.
> In the revised version, we will conduct a more thorough literature review and incorporate the relevant works you mentioned. Here, we briefly compare our approach with the key related works as follows:
> * In [1], the authors' goal is to recover the follower's payoff matrix. In this vein, they introduce "inducibility gap" to characterize the hardness that arises from some actions that can be rarely induced. However, here we only care about optimizing the leader's policy. And if some follower actions can be rarely induced no matter what policy the leader is taking, the leader does not need to learn much about that action for this purpose.
> * In [2] and [3], the authors assume the follower uses best reponse model instead of quantal response. Therefore, the problem is ill-posed so all of their results have exponential dependence on some parameters in the worst case.
> To see the reason why the best response model has such bad behavior, please refer to the general response.
> * In [4], the authors assume the follower is bounded rational and construct a quantal response model, which is the same ours. However, their work mainly focus on *the computation problem*, which is how to compute Stackelberg equilibrium when the model (i.e. the follower and the leader's reward functions in their setting) is known. In the mean while, our work mainly focus on *the learning problem*, which is how to learn Stackelberg equilibrium from online data when the model is unknown.
> Moreover, they mainly studied security game, a special Stackelberg game without state transitions, while our work incorporates general cases.
> Thus, although we share similar assumptions with theirs, the focuses and targets in these works are diversed.
>
> ### Q2: How far away are the proposed techniques from addressing the same problem but with a far-sighted follower? What are the main obstacles?
> For farsighted follower, according to the definition of the quantal response (copy Chen at el Eqn 3.1), the MLE loss for quantal response now begin
> $$
>   \mathcal{L} _ h^t(\theta,P) = - \sum_{i=1}^{t-1} \eta A _ h^{\pi^i, \theta, P},
> $$
> * The first challenge is the coupling between the follower's unknown reward parameter $\theta$ and the unknown transition kernel $P$. This coupling occurrs as the follower's cumulative reward admits a Bellman update with respect to the transition kernel.
> * The second challenge is that the advantage function need to be calculated with the recursion
> $$
> \begin{align*}
> A _ h^{\theta, P, \pi}(s_h, b_h) &= Q _ h^{\theta, P, \pi}(s _ h, b _ h) - V _ {h}^{\theta, P, \pi}(s _ h),\\\\
>   Q _ h^{\theta, P, \pi} (s _ h, b _ h) &= \\{ r _ h^{\theta, \pi} + \gamma  ( P _ h^\pi V _ {h+1}^{\theta, P, \pi} )\\}(s _ h, b _ h), \\\\
> V _ h^{\theta, P, \pi}(s _ h) &= \eta^{-1} \log \biggl( \sum _ {b\in \mathcal{B}} \exp\bigl( \eta Q _ h^{\theta, P, \pi}(s _ h, b _ h) \bigr) \biggr).
> \end{align*}
> $$
> This also poses significant challenges to optimization, as the optimization in the later steps can significantly influences the preceeding steps.
> * The last challenge is an $\exp(H)$ dependency in the regret bound (See the regret bound in Theorem 7.2 of Chen at el where $B_A$ is a constant that also depends on the Horizon $H$). How to get rid of this term still remains open.
>
> 1. Wu, Jibang, et al. "Inverse game theory for stackelberg games: the blessing of bounded rationality." Advances in Neural Information Processing Systems 35 (2022): 32186-32198.
> 2. Letchford, Joshua, Vincent Conitzer, and Kamesh Munagala. "Learning and approximating the optimal strategy to commit to." Algorithmic Game Theory: Second International Symposium, SAGT 2009, Paphos, Cyprus, October 18-20, 2009. Proceedings 2. Springer Berlin Heidelberg, 2009.
> 3. Peng, Binghui, et al. "Learning optimal strategies to commit to." Proceedings of the AAAI Conference on Artificial Intelligence. Vol. 33. No. 01. 2019.
> 4. Yang, Rong, Fernando Ordonez, and Milind Tambe. "Computing optimal strategy against quantal response in security games." AAMAS. 2012.
> 5. Yu Bai, Chi Jin, Huan Wang, and Caiming Xiong. Sample-efficient learning of stackelberg equilibria in general-sum games. Advances in Neural Information Processing Systems, 34:25799–25811,2021.

---

### Official Review · Reviewer_gWnr · 2024-11-10

**Soundness:** 2
**Presentation:** 2
**Contribution:** 2
**Rating:** 3
**Confidence:** 3

**Summary:**

This paper considers designing a policy for the leader in a Stackelberg game where the followers' reward function is unknown, requiring online learning.

The authors consider a setting with boundedly rational followers, where boundedly rational is defined as containing a regularization term, which is constrained to be strongly convex.

This paper introduces an algorithm called "Planning after estimation" (PES). PES follows two steps. For the first step, estimating the leaders' value function and the followers' quantal response model, the authors introduce a combined loss function that balances likelihood loss of the follower, Bellman loss of the leader, and an exploration term. For the second step, updating the leaders' policy, is done greedily.

The authors show that their approach achieves a sublinear $\tilde{\mathcal{O}} (d_c \sqrt{T})$ regret (where $d_c$ is a decoupling coefficient), and theoretically instantiate this with reward functions from RL with human feedback, used to train LLMs.

**Strengths:**

1. Stackelberg games are widely used across a variety of settings, and as the paper discusses, are applicable to RL with human feedback used to train LLMs. It is easy to justify the motivation of this paper, as in many real-world setting we would not know the followers' reward function a priori. Thus, realistically the leader would have to plan with imperfect information of the followers' response --- and learning this policy needs to be done in a computationally tractable (sample efficient) way.

2. The authors formulate an approach for solving online Stackelberg games with bilevel optimization, first estimating the followers' response and the leaders' own value function, then greedily planning after these estimates. Their approach contrasts to Chen et al. [2023] which instead builds confidence sets over the followers' response function, which is more computationally complex as the constrained optimization problem requires solving coupled confidencee sets.

3. The authors include analysis to show their approach, under certain assumptions, leads to a  small decoupling coefficient and achieves $\mathcal{O} (\sqrt{T})$.

**Weaknesses:**

It appears that this paper modifies the text size of the title (it looks vertically compressed), which is against submission policy.

There are no empirical results validating this approach; the "case study" just instantiates the reward functions. Given that one of the biggest benefits that the authors mention is that their method is more computationally tractable than Chen et al., this is a major weakness.

It is therefore not possible to determine whether this approach works in practice, or is just a hypothetical framework.



Specific comments

Writing could be improved in some points for easier reading.
- line 15: "posing" -> "which poses
- line 95: "in specific" -> "specifically"
- line 98: "omnisicent" -- as in rational and deterministic?
- paragraph 116-127: several things are capitalized ("Our" and "Multi") which should not be.
- line 132: "the set of probability measure on X" awkward wording
- line 114: "denotes" -> denote
- line 151: "step any step" extra word

**Questions:**

Writing could be improved in some points for easier reading:
- line 15: "posing" -> "which poses
- line 95: "in specific" -> "specifically"
- line 98: "omnisicent" -- as in rational and deterministic?
- paragraph 116-127: several things are capitalized ("Our" and "Multi") which should not be.
- line 132: "the set of probability measure on X" awkward wording
- line 114: "denotes" -> denote
- line 151: "step any step" extra wrod



Questions

1. Line 55: Why would the problem be ill-posed when the follower is fully rational? Shouldn't the problem be easier under perfect rationality?

2. Line 63: You say that Chen et al.'s approach for optimistic planning over confidence intervals considers a problem that is highly intractable --- but then how did they solve it? If the problem is very limited, e.g., only scales to a small problem size, it would be helpful to specify, and specify in which way your work overcomes these barriers.

3. Line 114: "Our algorithm is not only easy-to-implement but also easier to show theoretical guarantee" --> as in the math is easier? what does this mean?

4. Model: the transitions $P_h$ and reward function $u_h$ and $r_h$ suggest that the transitions and reward functions change at every timestep. Why is that the case? Shouldn't the transitions and reward functions be stationary? How does this evolution happen?

---

> ### Author Response · Authors · 2024-11-30
>
> We thank the reviewer for taking the time to read our paper, and for the comments and suggestions. Here are our main responses:
> ## Question about the submission policy.
> We would like to clarify that we strictly adhered to the conference's formatting guidelines during the preparation of our submission. The text size and formatting were applied as per the template provided by the conference, and no modifications were made beyond the specified requirements.
>
> If you believe there might still be an issue, we kindly suggest reporting this matter to the AC or PC for further verification. We are confident that our submission complies with the conference's formatting requirements, and we are happy to cooperate with any additional checks.
>
> ### Further explanation on "omnisicent".
> "omnisicent" is the word used by the authors in [1]. It means that they assume:
> 1. the follower knows the his/her reward function $\{r_h(s, a, b)\}_{h\in H}$ (For why we have a subindex $h$ in $r$, please see the answer to Q5);
> 2. the follower also knows the best response to any leader's action, i.e. for any $h \in [H]$, the mapping $f = \max_{b\in \mathcal{B}}r_h(s, a, b): \mathcal{A} \rightarrow \mathbb{R}$ is known;
> 3. the follower is myopic.
>
> We are sorry about the term abuse here and will add a short but precise explanation of this word in our revised paper accordingly.
>
> ### Q1: Why the problem is ill-posed when the follower is fully rational?
> Let's first recall the best response model of the myopic and fully rational follower:
> $$
> \upsilon_h^\pi (\cdot |s) = \arg\max_{\upsilon}\{\mathbb{E}_{a_h \sim \pi_h, b_h \sim \upsilon_h}[r_h(s_h, a_h, b_h) \mid s_h = s]\}.
> $$
> Intuitively speaking, this model is not robust to estimation errors in $r$, which means a slight estimation error in $r$ would result in a complete different best response action.
>
> Therefore, in this case, even if the leader could have a good estimation on the follower's reward function, the corresponding estimated response model may still be very different with the true model and such gap would corrupt our policy. For a rigorous and detailed discussion, please refer to Theorem 1 and Appendix C.1 in [2].
>
>
> ### Q2: Giving the fact that we state Chen et al's approach is highly intractable, how did they solve the problem? And what kind of specific barriers have we solved, compared with their approach?
>
> The key statement here is sample efficient algorithms are not necessarily computational tractable.
> In [3], they proposed a *sample efficient* algorithm to solve the Stackelberg equilbrium. It means we could achieve a given estimation error using finite samples (with polynomial dependence on the error tolerance and other parameters). However, such property does not mean we could calculated the solution in polynomial time.
>
> As what we have written in Line 336-359, Chen et al.'s approach requries to solve an *constraint optimization problem* in a complicated confidence set in planning steps. The planning steps have well-defined mathematical formulae, but it is very hard to calculate in practice.
> Instead, our approach only needs to solve an *unconstraint optimization problem* in planning steps, which reduces the computation costs.
>
> ### Q3: What does "...easier to show theoretical guarantee" mean?
> Specifically, here "easier" means that compared with [3], we circumvent the tedious confidence set construction and optimisitic planning step in our alrogithm.
> We shall use more precise words and statement in our revised paper.
>
> ### Q4: Shouldn't the transitions and reward functions be stationary? How does this evolution happen?
> To minimize potential understanding gaps, we assume the reviewer is referring to the transitions and reward functions being time-invariant.
>
> Our setting is a more general formulation, which could fully cover the stationary case by simply letting $P_h = P, r_h = r, \forall h \in [H]$. In our paper, we don't make any assumptions about the evolution of transitions and reward functions, and we allow policy to be non-stationary as well.
>
> 1. Geng Zhao, Banghua Zhu, Jiantao Jiao, and Michael Jordan. Online learning in stackelberg games with an omniscient follower. In International Conference on Machine Learning, pp. 42304–42316. PMLR, 2023.
>
> 2. Yu Bai, Chi Jin, Huan Wang, and Caiming Xiong. Sample-efficient learning of stackelberg equilibria in general-sum games. Advances in Neural Information Processing Systems, 34:25799–25811,
> 2021.
>
> 3. Siyu Chen, Mengdi Wang, and Zhuoran Yang. Actions speak what you want: Provably sampleefficient reinforcement learning of the quantal stackelberg equilibrium from strategic feedbacks.
> arXiv preprint arXiv:2307.14085, 2023.

---

### Author Response · Authors · 2024-11-30
**General Response**

Dear Reviewers,

We would like to thank all the reviewers for their thoughtful and constructive feedback on our work. Your comments have provided valuable insights that have helped us identify areas for improvement and refinement in our work.

In this general response, we aim to address several overarching concerns raised by multiple reviewers. For specific comments and detailed revisions, please refer to our point-by-point responses.

## Further illustration on agent response model:
In our work, we develop a learning algorithm from the leader's perspective, trying to find the optimal policy (maximize the cumulative reward) for the leader.

In Stackelberg game's structure, the leader's reward is not only depend on the state $s$ and his/her own action $a$, but also on the follower's action $b$. Thus, we need to model how would the follower react to the environment and the announced policy of the leader.

If we use the best response model to formulate the (myopic) follower's behavior, then we assume the follower is maximizing his/her reward for any given environment state and policy of the leader:
$$
\upsilon_h^\pi (\cdot |s) = \arg\max_{\upsilon}\{\mathbb{E}_{a_h \sim \pi_h, b_h \sim \upsilon_h}[r_h(s_h, a_h, b_h) \mid s_h = s]\}.
$$

If we use the the quantal response model to formulate the (myopic) follower's behavior, then we assume the follower is maximizing his/her reward with a regularized function for any given environment state and policy of the leader:
$$
\upsilon_h^\pi (\cdot |s) = \arg\max_{\upsilon}\{\mathbb{E}_{a_h \sim \pi_h, b_h \sim \upsilon_h}[r_h(s_h, a_h, b_h) \mid s_h = s] + \frac{1}{\eta}\mathcal{H}(\upsilon(\cdot|s))\},
$$

where $\mathcal{H}(\cdot)$ is the Shannon entropy and $\eta$ is the regularized coefficient.
Or equivalently, we could write the follower's response policy as
$$
\upsilon_h^{\pi}(b_h|s_h) = \frac{\exp(\eta\cdot (r_h^{\pi}(s_h, b_h)))}{\sum_{b_h \in B}\exp(\eta\cdot (r_h^{\pi}(s_h, b_h)))}
$$
From this formula, we could see the bounded rationality could be intepreted as: the follower will not take the best response, instead he/she will randomized his/her response with exponential weighting.

Different choices of $\eta$ (or even $\mathcal{H}(\cdot)$) represent different follower's behavior. In our setting, we assume the follower have such knowledge of the follower.

At last, we would like to strengthen that quantal response model and bounded rationality are extensively used in the context of Stackelberg Games([1], [2], [3]) and real-world applications ([4], [5]), which could partly prove that such models are reasonable and meaningful.

## Why the best response model is ill-posed and hard to use?
Intuitively speaking, this model is not robust to estimation errors in $r$, which means a slight estimation error in $r$ would result in a complete different best response action.

Therefore, in this case, even if the leader could have a good estimation on the follower's reward function, the corresponding estimated response model may still be very different with the true model and such gap would corrupt our policy. For a rigorous and detailed discussion, please refer to Theorem 1 and Appendix C.1 in [6].

We appreciate the opportunity to further clarify and improve our work, and we look forward to any additional feedback you may have after reviewing our responses and revisions.

1. Černý, Jakub, et al. "Computing quantal stackelberg equilibrium in extensive-form games." Proceedings of the AAAI Conference on Artificial Intelligence. Vol. 35. No. 6. 2021.

 2. Rong Yang, Fernando Ordonez, and Milind Tambe. Computing optimal strategy against quantal response in security games. In AAMAS, pages 847–854, 2012.

3. Arunesh Sinha, Debarun Kar, and Milind Tambe. Learning adversary behavior in security games: A pac model perspective. arXiv preprint arXiv:1511.00043, 2015.

4. Fei Fang, Thanh H Nguyen, Rob Pickles, Wai Y Lam, Gopalasamy R Clements, Bo An, Amandeep Singh, Brian C Schwedock, Milin Tambe, and Andrew Lemieux. Paws—a deployed game-theoretic application to combat poaching. AI Magazine, 38(1):23–36, 2017.

5. Bo An, Fernando Ordóñez, Milind Tambe, Eric Shieh, Rong Yang, Craig Baldwin, Joseph DiRenzo III, Kathryn Moretti, Ben Maule, and Garrett Meyer. A deployed quantal response- based patrol planning system for the us coast guard. Interfaces, 43(5):400–420, 2013.

6. Yu Bai, Chi Jin, Huan Wang, and Caiming Xiong. Sample-efficient learning of stackelberg equilibria in general-sum games. Advances in Neural Information Processing Systems, 34:25799–25811,2021.

---

### Meta-Review · Area_Chair_RNB7 · 2024-12-18

**Metareview:**

This paper examines the question of learning in episodic, two-player Markov Stackelberg games (MSGs) between a leader and a single follower. The authors assume that the follower is reacting to the leader's policy at each episode using a quantal response - actually, a *logit* response - policy, and they propose an algorithm for the leader, “Planning after Estimation” (PES), which jointly estimates the leader’s value function and the follower’s response mapping by optimizing a combined loss function; the algorithm subsequently updates the leader’s policy through a greedy "planning" step. The authors then show that, with high probability, the proposed algorithm guarantees that the leader's regret is at most $\mathcal{O}(\sqrt{T})$.

The reviewers appreciated the author's work on a challenging question and found the paper relatively well-written (at least, for the most part). At the same time, the review and discussion process also brought to the forefront several limitations, namely the potential intractability of the function approximation approach, the knowledge and information requirements of the proposed framework (namely that both the leader and the follower are aware of each other's policy at each stage of the process), and the fact that, given prior related work, the authors' contribution seems somewhat incremental.

These issues were all raised during the reviewing phase, but the authors' replies did not sway the reviewers' opinion and assessment. In the end, it was not possible to make a clear case for acceptance, so a decision was reached to make a reject recommendation.

**Additional Comments On Reviewer Discussion:**

To conform to ICLR policy, I am repeating here the relevant part of the metareview considering the reviewer discussion.

> The review and discussion process brought to the forefront several limitations, namely the potential intractability of the function approximation approach, the knowledge and information requirements of the proposed framework (namely that both the leader and the follower are aware of each other's policy at each stage of the process), and the fact that, given prior related work, the authors' contribution seems somewhat incremental.
>
> These issues were all raised during the reviewing phase, but the authors' replies did not sway the reviewers' opinion and assessment.

---

### Decision · Program_Chairs · 2025-01-22

Reject